# Dop1 enhances conspecific olfactory attraction by inhibiting miR-9a maturation in locusts

Xiaojiao Guo[1,2], Zongyuan Ma[1,2], Baozhen Du[2], Ting Li[2], Wudi Li[1], Lingling Xu[1], Jing He[1] & Le Kang[1,2]

Dopamine receptor 1 (Dop1) mediates locust attraction behaviors, however, the mechanism by which Dop1 modulates this process remains unknown to date. Here, we identify differentially expressed small RNAs associated with locust olfactory attraction after activating and inhibiting Dop1. Small RNA transcriptome analysis and qPCR validation reveal that Dop1 activation and inhibition downregulates and upregulates microRNA-9a (miR-9a) expression, respectively. miR-9a knockdown in solitarious locusts increases their attraction to gregarious volatiles, whereas miR-9a overexpression in gregarious locusts reduces olfactory attraction. Moreover, miR-9a directly targets *adenylyl cyclase 2* (*ac2*), causing its downregulation at the mRNA and protein levels. *ac2* responds to Dop1 and mediates locust olfactory attraction. Mechanistically, Dop1 inhibits miR-9a expression through inducing the dissociation of La protein from pre-miR-9a and resulting in miR-9a maturation inhibition. Our results reveal a Dop1–miR-9a–AC2 circuit that modulates locust olfactory attraction underlying aggregation. This study suggests that miRNAs act as key messengers in the GPCR signaling.

[1] State Key Laboratory of Integrated Management of Pest Insects and Rodents, Institute of Zoology, Chinese Academy of Sciences, Beijing 100101, China.
[2] Beijing Institutes of Life Sciences, Chinese Academy of Sciences, Beijing 100101, China. These authors contributed equally: Xiaojiao Guo, Zongyuan Ma. Correspondence and requests for materials should be addressed to L.K. (email: lkang@ioz.ac.cn)

Dopamine, a conservative and effective neurotransmitter in vertebrates and invertebrates, plays important roles in the regulation of diverse behaviors. This neurotransmitter modulates animal behaviors at different levels, including its synthesis[1], transport and release[2], reception and signaling[3,4], and neuron signals[5,6]. Numerous studies in mammals focused on dopamine receptors and related signaling because of their specific recognition to dopamine and their key roles in initiating downstream cascades[7,8]. Five dopamine receptor subtypes have been identified in vertebrates and are classified into two major groups, namely, D1-like (D1 and D5) and D2-like (D2, D3, and D4) receptors[8]. Signaling through dopamine receptors regulates motor activity[9], motivation and reward[10], social cognition[11], and various neurological and psychiatric diseases[12]. Invertebrates, specifically insects, have four subtypes of dopamine receptors, namely, D1-like dopamine receptor (Dop1), invertebrate-type dopamine receptors (INDRs or Dop2), D2-like dopamine receptor (Dop3), and DopEcR[13,14]. The functional characterization of dopamine receptors has been reported in a few species. DAMB and dDA1 in the fly and Dop1 in the cricket are involved in olfactory learning and memory[3,15,16]. However, the molecular mechanisms underlying the downstream signaling of dopamine receptors are largely unknown.

The migratory locust, *Locusta migratoria*, shows phase polyphenism in response to changes in local population density[17]. Locusts in gregarious phase aggregate together and form conspecific swarms or marching bands, whereas solitarious locusts are cryptic and tend to live individually[17,18]. Previous studies revealed that dopamine and related genes, which are involved in its synthesis, transit, and signaling modulated locust phase transition[19–21]. The concentration of dopamine and the expression levels of key genes in the dopamine metabolic pathway are upregulated in the gregarious locusts compared to the solitarious locusts, and Dop1 activation through injecting dopamine and its agonists induces the gregarious behavior in the migratory locust[19,20]. In addition, arena behavioral assay indicated that Dop1 activation in solitarious locusts induces the rapid behavioral change from avoidance to mutual attraction (including olfactory and visual cues)[19]. Recent studies have found that gregarious and solitarious locusts exhibit attraction and repulsion responses to the volatiles from the body and feces of gregarious locusts, respectively[22,23]. The compound composition includes phenylacetonitrile, benzaldehyde, guaiacol, phenol, aliphatic acids, and 2,3-butanediol, which have been identified as the main components of gregarious volatiles[24]. Behavioral assay results showed that the olfactory preference of the locusts to the natural mixture of these components can change significantly during phase transition[22]. Thus, the olfactory attraction plays important roles in aggregation and swarm formation of the migratory locust. However, the function of Dop1 in locust olfactory attraction and the mechanism by which Dop1 modulates this process remains unknown to date.

D1 receptor activates multiple signaling pathways through G proteins after binding to dopamine, including the canonical adenylyl cyclase (AC)-cAMP-PKA and PLC pathways, which are found from insects to mammals[8,25,26]. The non-canonical kinase

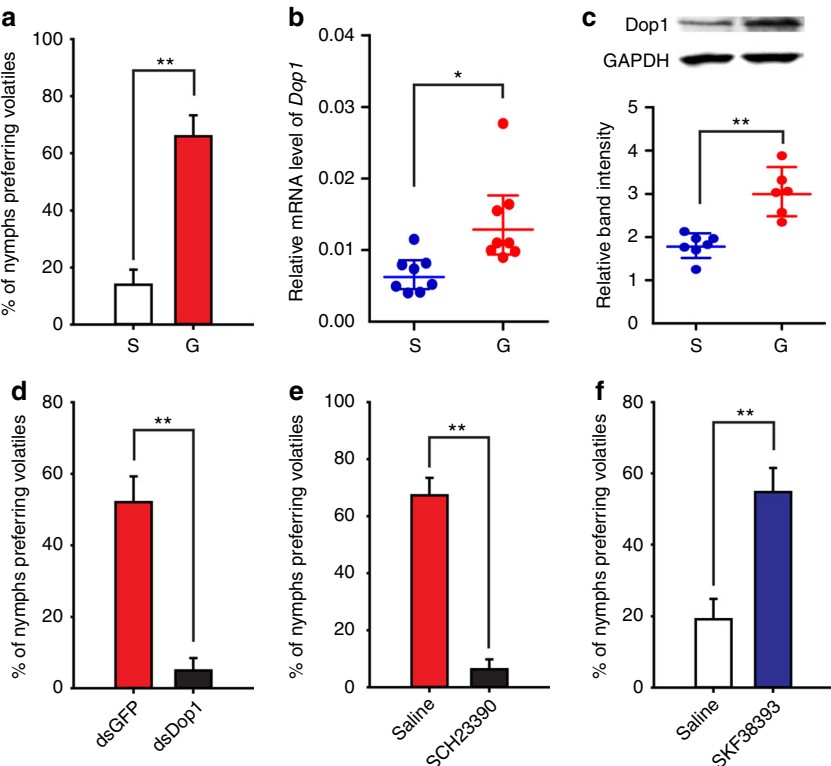

**Fig. 1** Dop1 regulates the olfactory attraction to gregarious volatiles in the migratory locust. **a** Gregarious locusts showed higher olfactory preference for gregarious volatiles than solitarious locusts ($n = 44$ and 41, *G*-test). **b, c** Dop1 mRNA (**b**, $n = 8$) and protein levels (**c**, $n = 6$) significantly differed in the brains of solitarious and gregarious locusts (Student's *t*-test). **d** Dop1 RNAi knockdown reduced the olfactory response of gregarious locusts to their volatiles ($n = 48$ and 40). **e** Injection of the Dop1 antagonist SCH23390 reduced the olfactory response of gregarious locusts to their volatiles ($n = 54$ and 48). **f** Injection of the Dop1 agonist SKF38393 enhanced the olfactory preference of solitarious locusts to gregarious volatiles ($n = 47$ and 41). The asterisks outside the strip indicate the significant difference between controls and the treatments through *G*-test for independence (**d**, **f**). The data in **b** and **c** are shown as mean ± SEM and the data in **a**, **d**, **e**, and **f** are shown as proportion (*p*) ± SE *$P < 0.05$; **$P < 0.01$. S: solitarious, G: gregarious

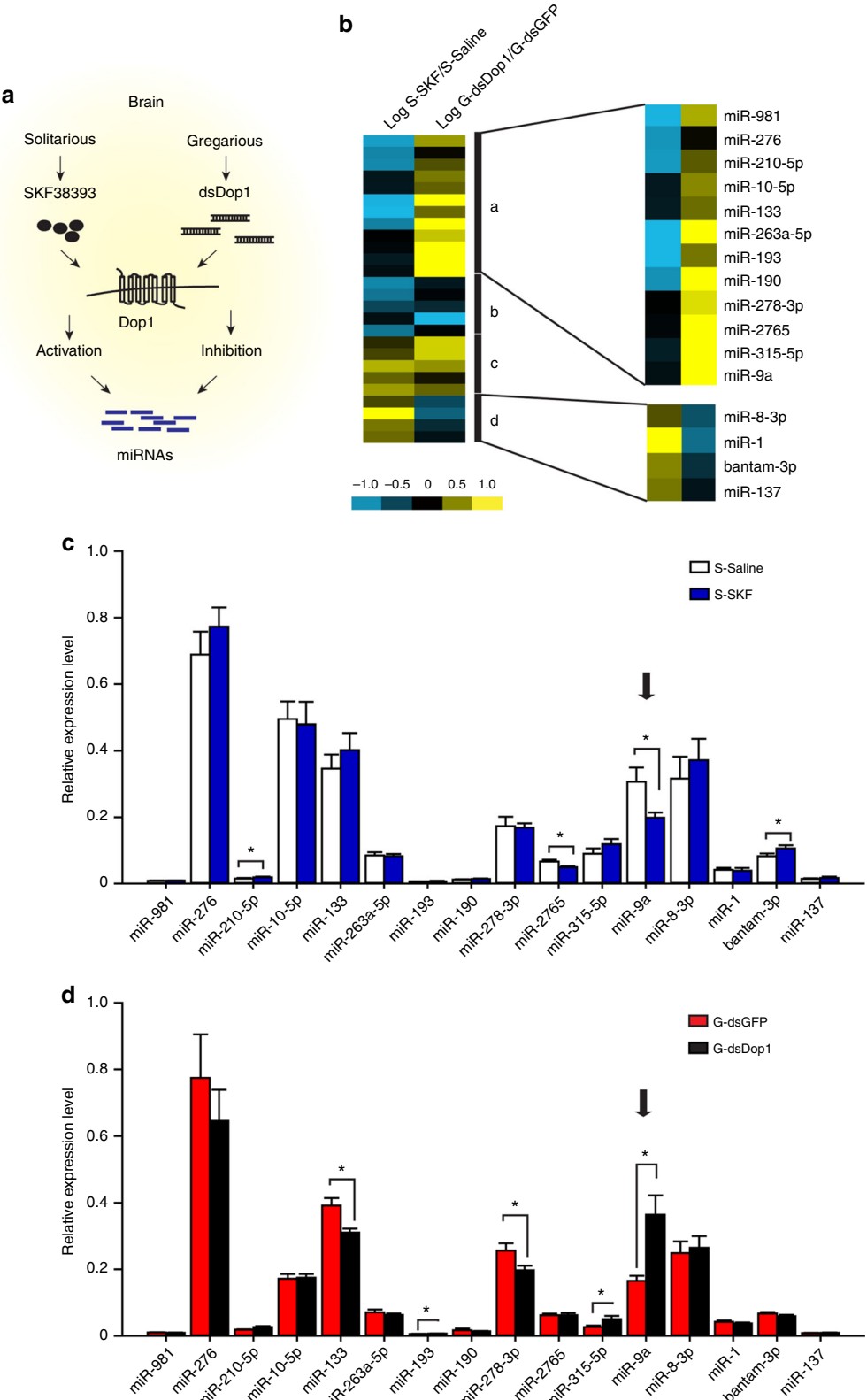

**Fig. 2** miR-9a responds to Dop1 administration in the brain of the migratory locust. **a** RNA-seq for miRNA expression profiles of the samples from solitarious brains with Dop1 agonist injection and gregarious brains with Dop1 RNAi knockdown. **b** Hierarchical clustering analysis of the differentially expressed miRNAs after activation and inhibition of Dop1. **c**, **d** Expression levels of differentially expressed miRNAs after SKF injection (**c**) and dsDop1 injection (**d**) as determined by qPCR. The asterisks outside the strip indicate the significant difference between the comparison of controls and the treatments through Student's *t*-test and presented as the mean ± SEM (*n* = 8); *P < 0.05. S: solitarious, G: gregarious, SKF: SKF38393

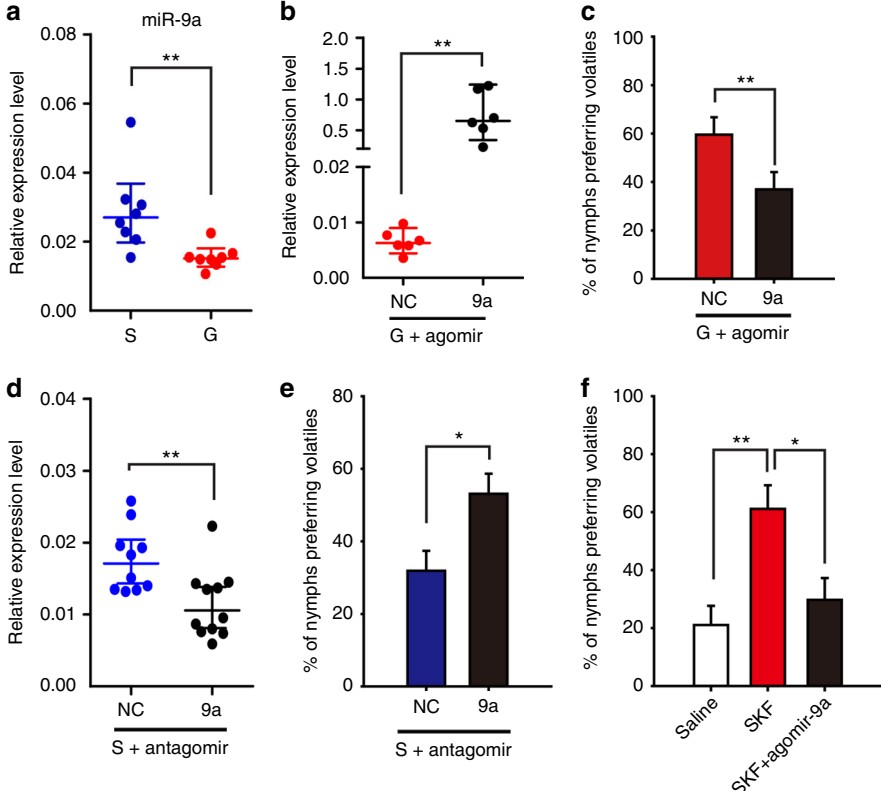

**Fig. 3** miR-9a mediates the olfactory preference for gregarious volatiles in the migratory locust. **a** Expression level of miR-9a in the brains of the solitarious and gregarious locusts ($n = 8$, Student's $t$-test). **b** Overexpression of miR-9a in the brains of gregarious locusts after injecting miR-9a agomir ($n = 6$). **c** Overexpression of miR-9a inhibited the olfactory preference of the gregarious to gregarious volatiles ($n = 41$ and 46). **d** Knockdown of miR-9a in the brains of solitarious locusts after injecting miR-9a antagomir ($n = 10$). **e** Knockdown of miR-9a enhanced the olfactory preference in the solitarious locusts ($n = 49$ and 43). **f** Administration of agomir-9a inhibited the olfactory preference of the solitarious locusts injected with SKF38393 ($n = 44, 48$, and 41). The asterisks outside the strip indicate the significant difference between controls and the treatments through Student's $t$-test (**b**, **d**) and $G$-test for independence (**c**, **f**); The data in **a**, **b**, and **d** are shown as mean ± SEM, and the data in **c**, **e**, and **f** are shown as $p$ ± SE. *$P < 0.05$; **$P < 0.01$. NC: negative control, S: solitarious, G: gregarious, SKF: SKF38393

Akt and ERK pathways are also regulated by dopamine receptors in different cells[8]. In addition, D1 receptor in mammals is a critical mediator for inducing the expression of immediate early genes, including transcription factors and neuropeptides[27,28]. Beyond the signaling cascade and coding genes, a recent study reported that the expression of a microRNA (miRNA) miR-181a was regulated by D1 receptor agonist in hippocampal neurons[29], suggesting that miRNAs are involved in the D1 signaling pathway. MiRNAs have emerged as important post-transcriptional regulators of neural development, physiology, and plasticity in animals[30,31]. Previous studies indicated the important roles of miRNAs in diverse biological processes of the migratory locust, including aggregation behavior[21], egg-hatching synchrony[32], and molting[33]. Notably, miRNA-133 can inhibit locust aggregation through targeting *henna* and *pale*, the two key genes in the dopamine synthesis pathway[21]. However, the mechanism by which Dop1 modulates the downstream miRNAs involved in locust olfactory attraction remains to be elucidated.

In the present study, we conducted activation and inhibition experiments of Dop1 to identify small RNAs associated with the olfactory behavior of the migratory locust. Results demonstrated that Dop1 modulates locust olfactory attraction by inhibiting the expression of miR-9a, the miRNA directly targeting *adenylyl cyclase 2* (*ac2*). In terms of mechanism, Dop1 inhibits miR-9a maturation by preventing the direct binding of the La protein to miR-9a precursor (pre-miR-9a). Our results reveal a molecular mechanism by which Dop1 regulates the downstream miRNAs and genes underlying locust olfactory behavior.

## Results

**Dop1 enhances locust attraction to gregarious volatiles**. We assessed olfactory preferences by giving each locust a single choice between gregarious volatiles and air in a Y-maze (Supplementary Figure 1) throughout this study. In this paradigm, gregarious locusts showed significantly higher preference to their own volatiles, compared with solitarious locusts ($G$-test, $G_2 = 14.64$, $P < 0.001$) (Fig. 1a). To determine whether the gregarious volatiles are the specific olfactory stimuli in the locust attraction, we tested the responses of gregarious locusts to the volatiles from other insect species. The gregarious locusts showed repulsion to the volatiles of *Solenopsis invicta* ($G$ test, $G_1 = 4.19$, $P = 0.020$), and showed no preference for the volatiles of *Acheta domesticus* ($G_1 = 0.24$, $P = 0.313$), *Blattella germanica* ($G_1 = 0.01$, $P = 0.5$), *Apis mellifera* ($G_1 = 0.01$, $P = 0.5$), and *Helicoverpa armigera* ($G_1 = 1.23$, $P = 0.133$), when the locusts were presented with volatiles and air (Supplementary Figure 2A). After giving the volatiles of gregarious locusts and other insect species, the gregarious locusts preferred conspecific gregarious volatiles to the volatiles of *Solenopsis invicta* ($G_1 = 13.95$, $P < 0.001$), *Acheta domesticus* ($G_1 = 5.82$, $P = 0.007$), *Blattella germanica* ($G_1 = 3.96$, $P = 0.023$), *Apis mellifera* ($G_1 = 3.96$, $P = 0.023$), and *Helicoverpa armigera* ($G_1 = 7.46$, $P = 0.003$) (Supplementary Figure 2B). Thus, the migratory

locust shows specific olfactory attraction to volatiles of conspecific gregarious locusts, but not general to an olfactory stimulus.

The mRNA and protein levels of Dop1 were significantly higher in the brains of gregarious locusts than in those of solitarious ones (Student's $t$-test, $t = 2.633$, $P = 0.025$ for mRNA;

$t = 4.765$, $P = 0.001$ for protein) (Fig. 1b, c). The blockade of Dop1 in gregarious locusts through RNAi knockdown and SCH23390 (a specific antagonist[34,35]) injection significantly reduced their attraction to gregarious volatiles (RNAi knockdown, $G_2 = 27.74$, $P < 0.001$; SCH23390, $G_2 = 66.63$, $P < 0.001$)

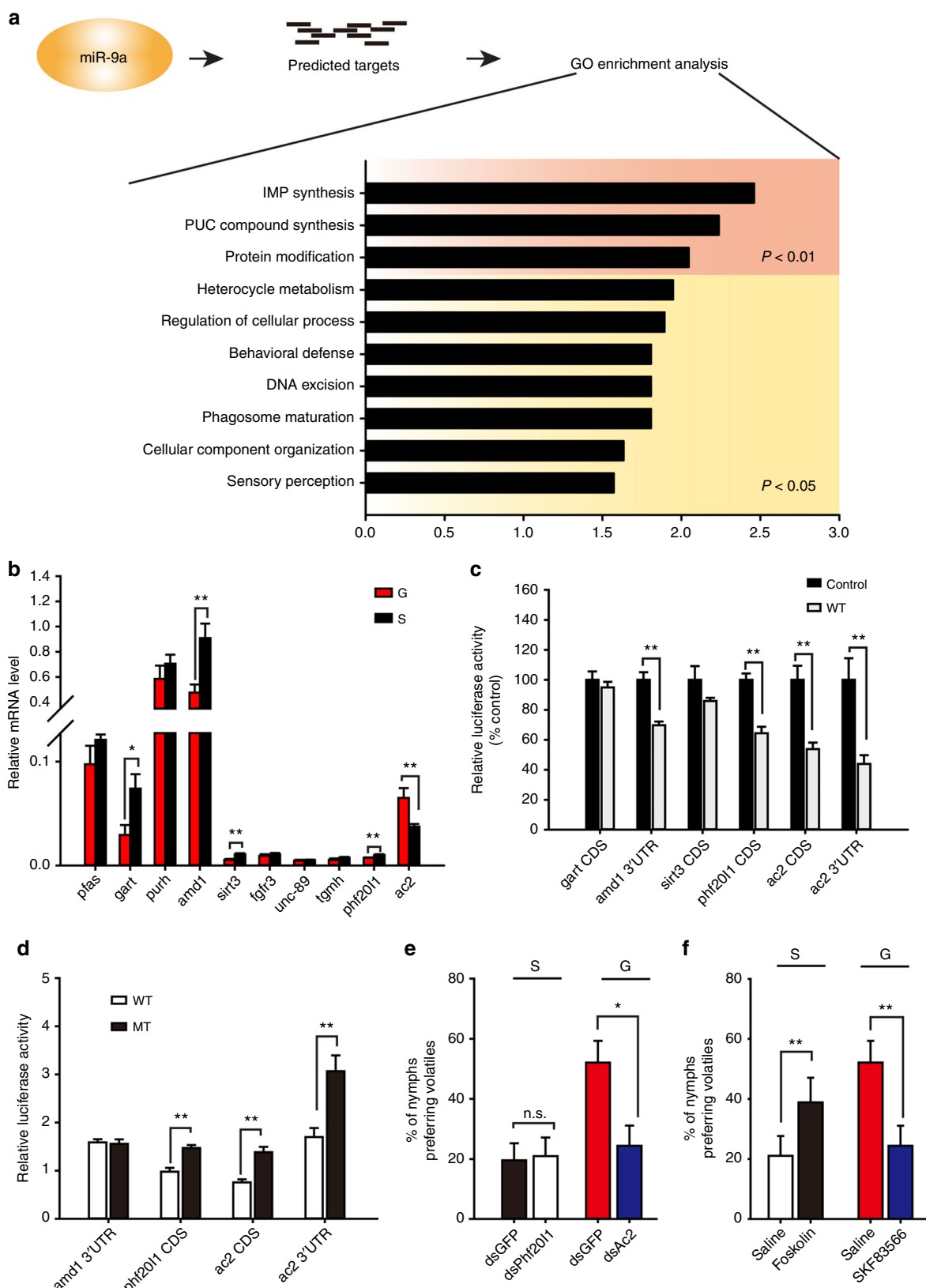

(Fig. 1d, e; Supplementary Figure 3). By contrast, injecting the specific agonist SKF38393 (SKF)[35,36] of Dop1 in the brains of solitarious locusts significantly induced the olfactory attraction to gregarious volatiles ($G_2 = 18.41$, $P < 0.001$) (Fig. 1f). Thus, Dop1 is linked with locust olfactory attraction to gregarious volatiles.

**Dop1–regulated miR-9a inhibits locust olfactory attraction**. Using small RNA transcriptome analysis, we analyzed the expression profile of miRNAs in the solitarious brains after Dop1 activation, and in the gregarious brains after Dop1 inhibition to determine the potential miRNAs associated with Dop1 (Fig. 2a). The migratory locust genome[37] was used to identify miRNAs in the locusts, and we found 661, 536, 561, and 565 miRNAs in the brain tissues injected with dsGFP, dsDop1, saline, and SKF38393, respectively (Supplementary Figures 4 and 5A). The locust miRNA homologs were identified using all known arthropod miRNAs released in the miRbase by homology searches. A total of 50 putative miRNA precursors, including 5P-miRNAs and/or their corresponding 3P-miRNAs, showed significant similarity to at least one of the known arthropod miRNAs (Supplementary Figure 5A). Among these conserved miRNAs expressed differentially, 9 were upregulated and 17 were downregulated in the SKF group (SKF vs. saline), whereas 17 were upregulated and 9 were downregulated in the dsDop1 group (dsDop1 vs. dsGFP) (Supplementary Figure 5B). Thermal map showed that these differentially expressed miRNAs were divided into four clusters, according to their similar or opposite expression patterns in SKF and dsDop1 groups (Fig. 2b). We applied quantitative polymerase chain reaction (qPCR) to validate the expression patterns of the miRNAs showing the opposite changes (clusters a and d) after the activation and inhibition of Dop1. The expression of miR-9a and miR-2765 significantly decreased ($t = 2.503$, $P = 0.026$ for miR-9a; $t = 2.498$, $P = 0.028$ for miR-2765), whereas bantam-3p and miR-210-5p significantly increased ($t = 2.602$, $P = 0.023$ for bantam-3p; $t = 2.624$, $P = 0.020$ for miR-210-5p) in the solitarious brains after SKF injection (Fig. 2c). In the gregarious brains injected with dsDop1, the expressions of miR-133 and miR-278-3p decreased significantly ($t = 3.217$, $P = 0.010$ for miR-133; $t = 2.308$, $P = 0.037$ for miR-278-3p), whereas miR-9a, miR-193, and miR-315-5p increased significantly ($t = 4.083$, $P = 0.002$ for miR-9a; $t = 2.501$, $P = 0.029$ for miR-193; $t = 2.650$, $P = 0.023$ for miR-315-5p) (Fig. 2d). The expression levels of miR-210-5p, miR-133, and miR-278-3p detected by RNA-seq and qPCR showed discrepancy. Based on the qPCR results, miR-9a showed significant and opposite changes in SKF and dsDop1 injection groups. Thus, miR-9a significantly responds to Dop1 administration in the brain of the migratory locust.

miR-9a showed a higher expression in the brains of solitarious locusts than in those of gregarious locusts ($t = 3.615$, $P = 0.002$) (Fig. 3a). To determine the role of miR-9a in the olfactory attraction of the migratory locust, we injected agomir-9a and antisense oligonucleotides (antagomir-9a) to promote and inhibit miR-9a expression in the locust brain, respectively. The expression level of miR-9a dramatically increased 48 h after

injecting 42 pmol of agomir-9a in the brains of gregarious locusts ($t = 4.935$, $P = 0.004$) (Fig. 3b). Behavioral assay in Y-maze showed that agomir-9a–injected gregarious locusts decreased the percentage of olfactory attraction for gregarious volatiles compared with the control group injected with agomir-NC (cel-miR-67-3p used as the negative control) ($G_2 = 12.714$, $P = 0.002$) (Fig. 3c). By contrast, the injection of antogomir-9a in the brains of solitarious locusts significantly decreased miR-9a expression ($t = 4.092$, $P = 0.001$) (Fig. 3d). Meanwhile, antagomir-9a injection increased the attractive percentage for gregarious volatiles compared with the control group injected with antagomir-NC ($G_2 = 7.03$, $P = 0.029$) (Fig. 3e). For determining whether Dop1–induced miR-9a reduction directly regulates locust olfactory preference, we conducted rescue experiments by administering agomir-9a to enhance miR-9a expression in the solitarious locusts injected with SKF38393. Agomir-9a administration resulted in the robust reversal of the olfactory preference in solitarious locusts ($G_2 = 7.87$, $P = 0.019$) (Fig. 3f). These results demonstrate that miR-9a is a key modulator regulated by Dop1 and mediates the olfactory attraction in the locusts.

**Adenylyl cyclase 2 mediates miR-9a function in attraction**. We employed the algorithms miRanda[38] and RNAhybrid[39] to predict the potential target genes of miR-9a in the gene database from locust brain transcriptome (*Locusta migratoria*) (Supplementary Data 1). Among these candidates, 157 genes showed significant differences between the gregarious and solitarious phases according to the transcriptome data (fold change > 1.5) (Supplementary Data 2). GO enrichment analysis of biological processes showed that 10 groups were enriched. Moreover, three groups, including IMP biosynthetic process, purine-containing compound biosynthetic process, and protein modification process, were significantly enriched ($P < 0.01$) (Fig. 4a). Among these clusters, 10 genes, including *phosphoribosylformylglycinamidine synthase* (*pfas*), *trifunctional purine biosynthetic protein adenosine-3* (*gart*), *bifunctional purine biosynthesis protein* (*purh*), *S-adenosylmethionine decarboxylase proenzyme* (*amd1*), *NAD-dependent deacetylase sirtuin-3* (*sirt3*), *fibroblast growth factor receptor 3* (*fgfr3*), *muscle M-line assembly protein unc-89* (*unc-89*), *hemocyte protein-glutamine gamma-glutamyltransferase* (*tgmh*), *PHD finger protein 20-like protein 1* (*phf20l1*), and *adenylyl cyclase 2* (*ac2*) were significantly enriched. qPCR validation showed that *gart*, *amd1*, *sirt3*, *phf20l1*, and *ac2* exhibited significant differences between the gregarious and solitarious phases ($t = 2.737$, $P = 0.016$ for *gart*; $t = 3.406$, $P = 0.005$ for *amd1*; $t = 4.099$, $P = 0.001$ for *sirt3*; $t = 3.087$, $P = 0.008$ for *phf20l1*; $t = 3.957$, $P = 0.005$ for *ac2*) (Fig. 4b).

The direct interactions between miR-9a and putative target genes (*gart*, *amd1*, *sirt3*, *phf20l1*, and *ac2*) in *Drosophila* S2 cells were verified by luciferase assays. miR-9a significantly reduced the luciferase activities of the constructs containing *amd1* 3′UTR, *phf20l1* CDS, *ac2* CDS, and 3′UTR target sites ($t = 5.357$, $P < 0.001$ for *amd1* CDS; $t = 5.838$, $P < 0.001$ for *phf20l1* CDS; $t = 4.293$, $P < 0.001$ for *ac2* CDS; $t = 4.707$, $P = 0.001$ for *ac2*

**Fig. 4** miR-9a–targeted *adenylyl cyclase 2* (*ac2*) mediates locust preference for gregarious volatiles. **a** Functional cluster of miR-9a target genes using GO enrichment analysis. **b** mRNA level of genes in the clusters ($P < 0.01$) as validated by qPCR ($n = 8$, Student's *t*-test). **c** Luciferase reporter assays in S2 cells co-transfected with miR-9a overexpression vectors and psiCHECK2 vectors containing binding sites of the target genes ($n = 6$, Student's *t*-test). **d** Luciferase reporter assays in S2 cells co-transfected with miR-9a overexpression vectors and psiCHECK2 vectors containing wild (WT) or mutant (MT) sequences of target genes ($n = 6$, Student's *t*-test). **e** Olfactory responses of the solitarious and gregarious locusts injected with dsPhf20l1 ($n = 46$ and 43) and dsAc2 ($n = 40$ and 46), respectively. **f** Olfactory responses of the solitarious and gregarious locusts injected with AC2 agonist (forskolin, $n = 44$ and 45) and antagonist (SKF83566, $n = 48$ and 41), respectively. The data in **b**, **c**, and **d** are shown as mean ± SEM, and the data in **e** and **f** are shown as $p$ ± SE. The asterisks outside the strip indicate the significant difference between controls and the treatments by *G*-test for independence (**e** and **f**); n.s.: not significant; *$P < 0.05$; **$P < 0.01$; S: solitarious, G: gregarious

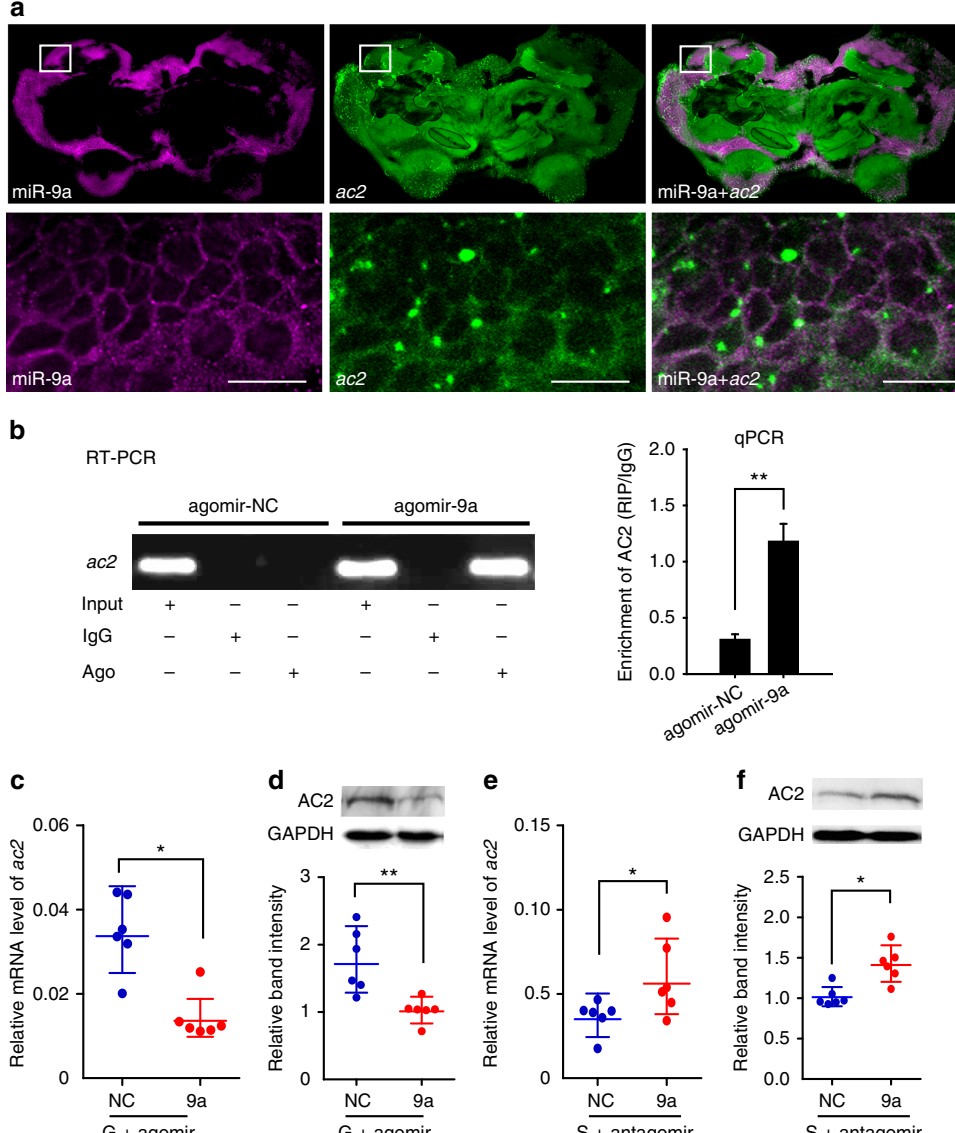

**Fig. 5** miR-9a inhibits *ac2* mRNA and protein expression by direct interaction. **a** Co-localization of miR-9a and *ac2* in the locust brain through the co-labeling of miRNA and mRNA with fluorescence in situ hybridization. The squares specifically indicate the areas where miR-9a and *ac2* were localized in the Kenyon cells of mushroom body. A white signal is observed in the area with magenta (miR-9a) and green signals (*ac2*) overlapping, indicating the co-localization of miR-9a and its target. The images were visualized using an LSM 710 confocal fluorescence microscope (Zeiss) at magnifications of ×10 (the upper row) and ×20 (the bottom row). Scale bar represents 20 μm. **b** RNA immunoprecipitation (RIP) was performed with an anti-Ago-1 antibody; normal mouse IgG was used as a negative control. RT-PCR or qPCR analysis was performed to amplify *ac2* mRNA from the Ago-1 immunoprecipitates in brain tissue extracts treated with agomir-9a compared with agomir-control (agomir-NC). **c, d** Relative mRNA level (**c**, $n = 6$) and protein level (**d**, $n = 6$) of *ac2* in the brains of the gregarious locusts injected with agomir-9a. **e, f** Relative mRNA level (**e**, $n = 6$) and protein levels (**f**, $n = 6$) of *ac2* in the brains of the solitarious locusts injected with antagomir-9a. The data for RIP assay, qPCR, and Western blot are presented as the mean ± SEM; The asterisks outside the strip indicate the significant difference between controls and the treatments by Student's *t*-test. *$P < 0.05$; **$P < 0.01$. S: solitarious, G: gregarious, NC: negative control

3'UTR) (Fig. 4c). The mutations in the binding site of target genes (Supplementary Figure 6) abolished the suppression effects of miR-9a on the reporters with target sites from *phf20l1* CDS, *ac2* CDS, and 3'UTR ($t = 5.050$, $P < 0.001$ for *phf20l1* CDS; $t = 4.906$, $P < 0.001$ for *ac2* CDS; $t = 3.611$, $P = 0.004$ for *ac2* 3'UTR) (Fig. 4d). Thus, *phf20l1* and *ac2* are the putative target genes of miR-9a.

To determine the functions of *phf20l1* and *ac2* in the locust olfactory responses, we injected dsRNA of *phf20l1* and *ac2* into the brain of solitarious and gregarious locusts, respectively. At 72 h after the injection, *ac2* knockdown ($t = 2.677$, $P = 0.023$) (Supplementary Figure 7A) in the brains of gregarious locusts

significantly reduced their olfactory attraction to gregarious volatiles ($G_2 = 9.16$, $P = 0.010$), whereas *phf20l1* knockdown ($t = 3.023$, $P = 0.009$) (Supplementary Figure 7B) did not affect the olfactory responses of solitarious locusts ($G_2 = 0.59$, $P = 0.744$) (Fig. 4e). In addition, we detected the olfactory responses of gregarious locusts after *phf20l1* knockdown ($t = 3.818$, $P = 0.002$) (Supplementary Figure 8A), and we found that *phf20l1* knockdown did not change the olfactory preference of gregarious locusts ($G_2 = 0.23$, $P = 0.892$) (Supplementary Figure 8B). These results indicate that *ac2*, but not *phf20l1*, is involved in locust olfactory preference.

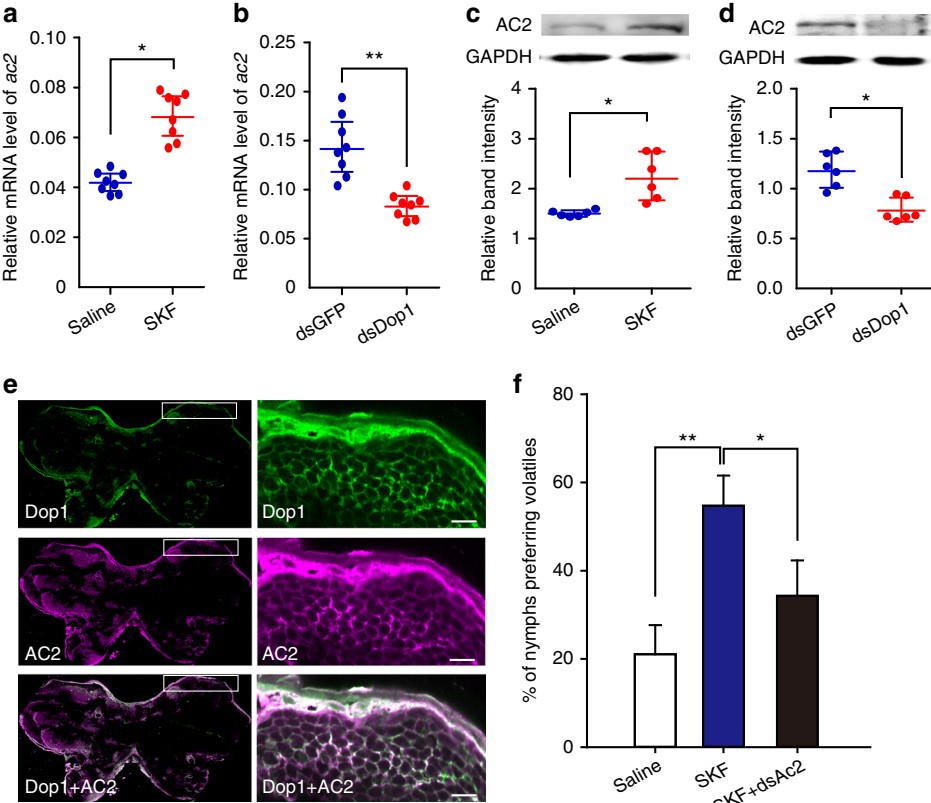

**Fig. 6** *ac2* responds to Dop1 administration in the brain of migratory locust. **a**, **b** The qPCR analysis showed that the mRNA level of *ac2* was upregulated by SKF injection in the solitarious brains (**a**, $n = 8$) but downregulated by dsDop1 injection in the gregarious brains (**b**, $n = 8$). **c**, **d** Western blot analysis showed that the protein level of AC2 was upregulated by SKF injection in the solitarious brains (**c**, $n = 6$) but downregulated by dsDop1 injection in the gregarious brains (**d**, $n = 6$). **e** Co-localization of Dop1 and AC2 in the locust brain based on the immunofluorescence. The squares specifically indicate the areas where Dop1 and AC2 were localized in the Kenyon cells of the locust mushroom body. A white signal is observed in the area with green (Dop1) and magenta signals (AC2) overlapping, indicating the co-localization of Dop1 and AC2. The images were visualized using an LSM 710 confocal fluorescence microscope (Zeiss) at magnifications of ×10 (the left column) and ×20 (the right column). Scale bar represents 20 μm. **f** RNAi knockdown of *ac2* inhibited the olfactory preference of the solitarious locusts injected with SKF38393 ($n = 44$, 40, and 44). The data in **a** to **d** are shown as mean ± SEM and the data in **f** are shown as $p$ ± SE. The asterisks outside the strip indicate the significant difference between controls and the treatments through Student's *t*-test and *G*-test for independence. *$P < 0.05$. **$P < 0.01$. S: solitarious, G: gregarious, SKF: SKF38393

Furthermore, we applied pharmacology intervention to validate the role of AC2 in locust olfactory responses. The percentage of solitarious locusts preferring gregarious volatiles significantly increased ($G_2 = 11.08$, $P = 0.004$) after injecting forskolin (the agonist of adenylyl cyclase)[40] in the brains of solitary locusts (Fig. 4f). In gregarious locusts, blocking AC2 through antagonist (SKF83566)[41] injection significantly reduced their olfactory preference for gregarious volatiles ($G_2 = 21.84$, $P < 0.001$) (Fig. 4f). Therefore, AC2 has a direct link to the olfactory preference of locusts for gregarious volatiles.

**miR-9a reduces *ac2* mRNA and protein by direct targeting.** The localization of miR-9a and *ac2* was detected by double labeling with fluorescence in situ hybridization (FISH) to confirm the interaction between miR-9a and *ac2* in the brain. miR-9a and *ac2* co-localized in the Kenyon cells of the mushroom body in the locust brain, the higher olfactory processing center[42] (Fig. 5a). The negative control showed no signal in the same part of the brain (Supplementary Figure 9). The co-localization of miR-9a and *ac2* suggests the direct interactions between miR-9a and *ac2*.

To probe the interaction between miR-9a and *ac2*, we next performed RNA immunoprecipitation (RIP) using a monoclonal antibody against the Ago1 protein[21]. Ago1-immunoprecipitated

RNAs from the brains treated with agomir-9a were significantly enriched for *ac2*, compared with control Ago1-immunoprecipitated RNAs treated with agomir-NC ($t = 5.268$, $P = 0.006$) (Fig. 5b).

We injected agomir-9a and antagomir-9a in the brains of gregarious and solitarious locusts, respectively, to investigate whether miR-9a inhibits *ac2* expression in the brains of the locusts. The results of qPCR and Western blot showed that the mRNA and protein levels of *ac2* significantly decreased in the gregarious locusts injected with agomir-9a as compared with the control group injected with agomir-NC ($t = 2.490$, $P = 0.032$ for mRNA; $t = 3.896$, $P = 0.008$ for protein) (Fig. 5c, d; Supplementary Figure 10). By contrast, the mRNA and protein levels of *ac2* significantly increased in the brains of the solitarious locusts injected with antagomir-9a ($t = 2.707$, $P = 0.019$ for mRNA; $t = 3.413$, $P = 0.014$ for protein) (Fig. 5e, f). Thus, miR-9a inhibits *ac2* expression at the mRNA and protein levels through direct interaction.

**ac2 responds to Dop1 administration in the brain of locust.** Considering that *ac2* is a target of Dop1–regulated miR-9a, we investigated whether *ac2* responds to Dop1 administration. qPCR

and Western blot analyses were used to test the mRNA and protein levels of *ac2* after activating and inhibiting Dop1, respectively. The expression level of *ac2* mRNA significantly increased after injecting SKF38393 in the solitarious locusts, but decreased after the RNAi knockdown of Dop1 in the gregarious locusts ($t = 2.950$, $P = 0.016$, SKF38393 vs. saline; $t = 9.731$, $P < 0.001$, dsDop1 vs. dsGFP) (Fig. 6a, b). Similarly, the protein level

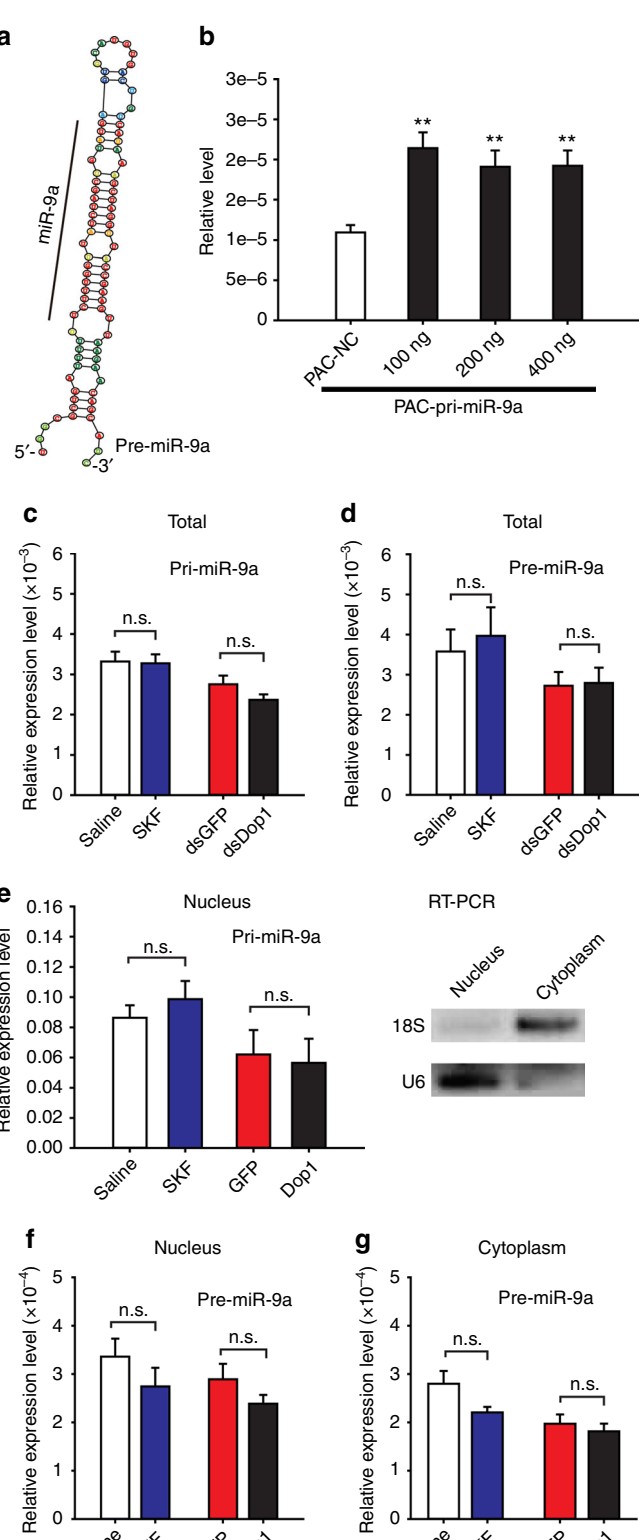

of AC2 significantly increased after SKF38393 injection but decreased after Dop1 dsRNA injection ($t = 3.166$, $P = 0.019$, SKF38393 vs. saline; $t = 3.025$, $P = 0.016$, dsDop1 vs. dsGFP) (Fig. 6c, d). Moreover, double-labeling immunohistochemistry in the brain showed that AC2 and Dop1 co-localized in the Kenyon cells of the locust mushroom body, suggesting the functional association of AC2 and Dop1 with olfactory processing (Fig. 6e). The negative control did not show any signal in the same part of the brain (Supplementary Figure 11). To determine whether the up-regulation of *ac2* induced by Dop1 directly regulates the locust olfactory preference, we conducted rescue experiments by injecting dsAc2 to reduce *ac2* expression in the solitarious locusts injected with SKF38393. RNAi knockdown abolished the increasing olfactory preference in the solitarious locusts induced by SKF38393 ($G_2 = 8.37$, $P = 0.015$) (Fig. 6f). Thus, Dop1 modulates *ac2* expression by regulating the miR-9a expression in the brain of the migratory locust.

**Dop1 inhibits miR-9a maturation through the La protein.** To determine the mechanism by which Dop1 modulates miR-9a expression, we cloned primary miR-9a (pri-miR-9a) and miR-9a precursor (pre-miR-9a) by referring to the genome and transcriptome database of the migratory locust on the basis of the biogenesis of miRNAs[43]. First, we constructed an expression vector with partial pri-miR-9a sequences, including pre-miR-9a sequences (Fig. 7a), and transfected the vector into S2 cells. The expression level of miR-9a in the transfected cells significantly increased as compared with that in the control group with mock transfection ($t = 4.822$, $P = 0.001$ for 100 ng; $t = 3.648$, $P = 0.004$ for 200 ng; $t = 3.932$, $P = 0.003$ for 400 ng) (Fig. 7b). This phenomenon indicates the sequence validity of pri-miR-9a and pre-miR-9a. We detected the expression levels of pri-miR-9a and pre-miR-9a in the brains of locusts after the activation and inhibition of Dop1 signaling, respectively. The expression levels of pri-miR-9a and pre-miR-9a were not affected by Dop1 ($P > 0.05$ for each group) (Fig. 7c, d). Pri-miRNA transcription and pre-miRNA processing occur in nucleus, and miRNAs mature occurs in cytoplasm[43]. Thus, to further reveal the effects of Dop1 on pri-miR-9a and pre-miR-9a levels at different cell components, we quantified the expression levels of pri-miR-9a and pre-miR-9a in the nucleus and cytoplasm by using qPCR. The expression of pri-miR-9a in the nucleus was not affected by Dop1 ($P > 0.05$ for each group) (Fig. 7e). The expression of pre-miR-9a in the nucleus and cytoplasm was also not affected by the activation and inhibition of Dop1, respectively ($P > 0.05$ for each group) (Fig. 7f, g). Considering that only the expression of mature miR-9a but not pri-miR-9a and pre-miR-9a is affected by Dop1, we speculate that Dop1 modulates the maturation from pre-miR-9a to miR-9a.

**Fig. 7** Dop1 does not affect the expression of primary miR-9a (pri-miR-9a) and miR-9a precursor (pre-miR-9a). **a** Predicted stem–loop structure of pre-miR-9a. **b** Expression of miR-9a in S2 cells transfected with expression vectors including partial sequence of pri-miR-9a centered pre-miR-9a ($n = 6$). **c**, **d** Expression levels of pri-miR-9a (**c**, $n = 8$) and pre-miR-9a (**d**, $n = 8$) in total cells after activation and inhibition of Dop1, respectively. **e** Expression levels of pri-miR-9a in the nucleus after activation and inhibition of Dop1, respectively. U6 was used as a nuclear marker and 18 S rRNA as a cytoplasmic marker. **f**, **g** Expression levels of pre-miR-9a in the nucleus (**f**, $n = 8$) and cytoplasm (**g**, $n = 8$) after activation and inhibition of Dop1, respectively. The asterisks outside the strip indicate the significant difference between controls and the treatments through Student's *t*-test and presented as the mean ± SEM. **$P < 0.01$. n.s.: not significant, SKF: SKF38393

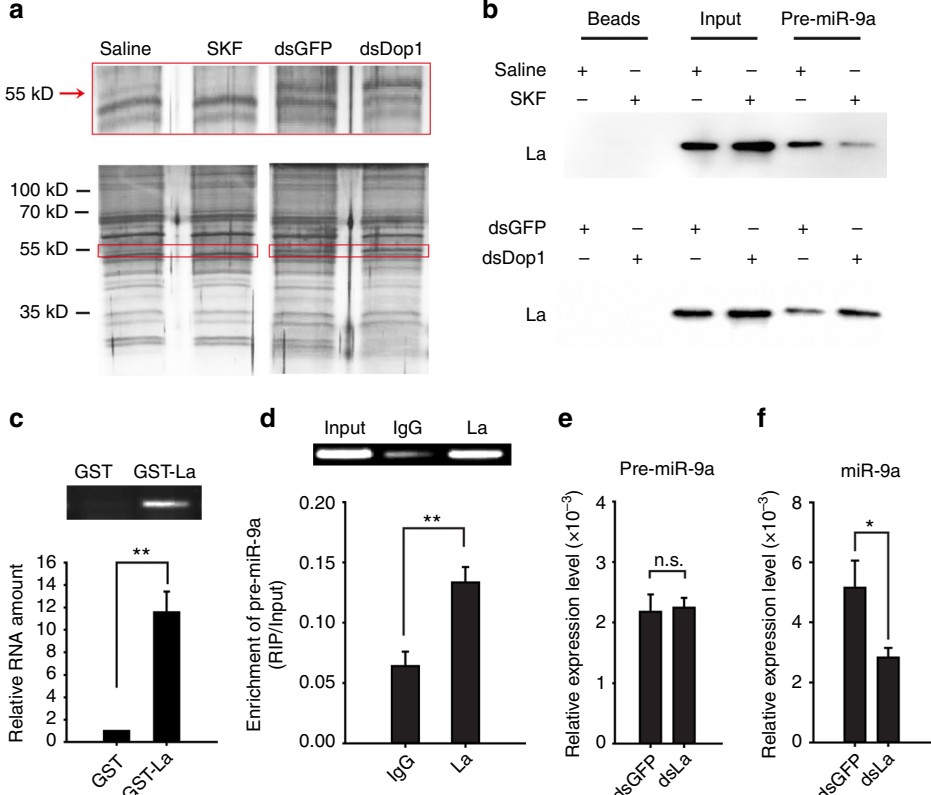

**Fig. 8** Dop1 inhibits miR-9a maturation by preventing the direct binding of the La protein with pre-miR-9a. **a** Pre-miR-9a RNA pull down in brain tissue extracts with Dop1 activation and RNAi knockdown. The 55 kD protein (red frame) showed the reversal change after Dop1 activation and knockdown. **b** Western blot analysis of the amount of La protein in RNA pull-down outputs from the brain tissue extracts treated with either Dop1 activation or RNAi knockdown. **c** RT-PCR and qPCR analyses of pre-miR-9a expression after pulling down with GST and GST–La fusion protein ($n = 4$). **d** RT-PCR and qPCR analyses of pre-miR-9a from La immunoprecipitates from brain tissue extracts ($n = 6$). **e, f** Expression levels of pre-miR-9a (**e**, $n = 6$) and miR-9a (**f**, $n = 6$) after knockdown of the La protein in the brain tissues. The data between controls and treatments were analyzed by Student's $t$-test and presented as the mean ± SEM. *$P < 0.05$; **$P < 0.01$. SKF: SKF38393

To further detect how Dop1 mediates the maturation of miR-9a, we used RNase-assisted RNA pull-down and mass spectrometry to identify the potential pre-miR-9a-binding proteins that are possibly modulated by Dop1. Biotin-labeled pre-miR-9a probe (Supplementary Table 1) was used to precipitate the proteins in the brain extracts derived from the solitarious locusts injected with SKF38393 and saline, and those from the gregarious locusts injected with dsDop1 and dsGFP. The abundance of a 55 kD protein decreased in the brains injected with SKF38393 but increased after dsDop1 injection (Fig. 8a). Mass spectrometry assay identified this protein as a putative La protein with the highest identify score (Supplementary Table 2), a RNA-binding protein for miRNA biogenesis[44].

Immunoblotting assay confirmed the existence of La protein in the protein complex extracted by RNA pull-down. To explore the functional association of the La protein with Dop1 signaling, we analyzed the abundance of this protein in the outputs after RNA pull-down. The abundance of La proteins that bind with pre-miR-9a decreased after Dop1 activation but increased after Dop1 RNAi knockdown (Fig. 8b; Supplementary Figure 12). Thus, Dop1 affects the binding of the La protein with pre-miR-9a. GST–La fusion protein pull-down was performed and pre-miR-9a was enriched in a RNA–protein complex (fold change >10) to further validate the interaction of the La protein with pre-miR-9a ($t = 6.783$, $P = 0.001$) (Fig. 8c). Moreover, RIP assay showed that the expression level of pre-miR-9a was significantly higher in the anti-La enrichments than that in the IgG controls, suggesting that the La protein directly binds with pre-miR-9a in vivo ($t = 3.951$, $P = 0.008$) (Fig. 8d). After knockdown of La protein by RNAi, the expression level of pre-miR-9a did not change, but that of miR-9a significantly decreased ($t = 0.212$, $P = 0.836$ for pre-miR-9a; $t = 2.428$, $P = 0.036$ for miR-9a) (Fig. 8e, f). Therefore, Dop1 inhibits the maturation of miR-9a by preventing the direct binding of La protein with pre-miR-9a.

## Discussion

In this study, we found that Dop1–inhibited miR-9a expression to modulate the olfactory attraction to gregarious volatiles in the migratory locust. *ac2* is a critical target of miR-9a in the regulation of locust olfactory attraction. Moreover, Dop1-inhibited miR-9a maturation by suppressing the binding of La protein with pre-miR-9a (see model in Fig. 9).

The function of dopamine in regulating aggregation behavior is closely correlated with epigenetic regulator miRNAs. Ma et al.[20] reported that dopamine and its related genes for synthesis (*pale* and *henna*), transport and release (*vat1*) were involved in the locust aggregation behavior. Yang et al.[21] found that miR-133 inhibits dopamine synthesis underlying behavioral aggregation through inhibiting two key genes, *pale* and *henna*. However, it is unknown whether and how dopamine mediates the expression of the miRNAs through dopamine receptors with downstream signal transduction system. Followed by the finding of Dop1–mediating locust aggregation behavior[19], we further found that Dop1 inhibits the maturation of miR-9a for olfactory

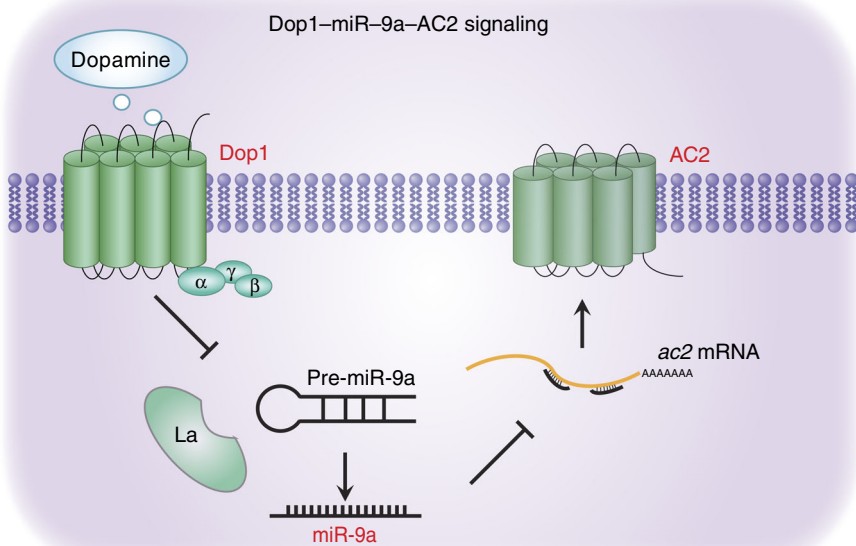

**Fig. 9** A model for Dop1 modulating AC2 expression through mediating the maturation of miR-9a. After binding to dopamine, Dop1 induces the disassociation of the La protein from pre-miR-9a and then the decrease in miR-9a expression. The downregulation of miR-9a abolishes its inhibiting effects on *ac2* and upregulates AC2 expression. The increase in AC2 finally induces locust olfactory preference for gregarious volatiles

behavioral regulation and the expression regulation of miR-133 is not correlated with Dop1 signaling. The novel mechanism that Dop1 modulates miR-9a maturation extends our understanding of dopamine action mode, and also provides a more comprehensive understanding of the function of dopamine in the migratory locust.

Activation of Dop1 in the brain induced locust olfactory attraction via the inhibition of miR-9a expression. The expressions of miRNAs are indeed modulated by different neurotransmitters and related agonists. miR-181a expression was induced by the D1 receptor agonist in hippocampal neuronal cells[29]. Activating the M3 subtype of muscarinic acetylcholine receptor (M3-mAChR) in H9c2 cells by using corresponding agonist promotes cardioprotection via the suppression of miR-376b-5p expression[45]. Adrenaline promotes cell proliferation and increases chemoresistance in colon cancer HT29 cells through the induction of miR-155 expression[46]. These findings together with ours suggest the regulation of miRNA expression by neurotransmitters is possibly widespread in different biological processes of animals.

Previous studies reported that the D1 receptor mediates gene expression at the transcriptional level through its downstream second messenger pathways. The D1 receptor in mammals is coupled with the activation of multiple transcription factors, including *zif-268*[28,47] and *jun-B*[28], at the mRNA level, and the cyclic AMP-response element binding protein (CREB) at the protein modification level[48]. Activating these transcription factors induces the transcription of downstream genes and results in the generation of nascent mRNAs. As a post-transcriptional regulator, miR-9a is regulated by Dop1, suggesting that Dop1 can modulate the downstream genes at the post-transcriptional level. Basing on these results, we speculated that the transcription of nascent mRNAs and the regulation of existing mRNAs levels are simultaneously mediated by Dop1, and the combined actions of these two levels are advantageous in regulating gene expression levels and controlling rapid behavioral changes. The findings of the present study extend the molecular mechanisms of dopamine receptors in regulating the expressions of downstream genes from the transcription level to the post-transcriptional level.

The mRNA and protein levels of *ac2* were regulated by Dop1 via the change in miR-9a level. As the key post-transcriptional regulator, miRNA functions as a guide by base pairing with its target mRNAs, whereas AGO proteins function as effectors by recruiting the factors that induce translational repression, mRNA deadenylation, and mRNA decay[49]. The reduced mRNA and protein levels of *ac2* may be attributed to the miR-9a-induced mRNA degradation and inhibition of translation. The change in *ac2* expression in response to Dop1 suggests that the dopamine receptor regulates the expression level of adenylyl cyclase. In the canonical dopamine pathway, the D1 dopamine receptor activates the catalytic activity of AC by G subunit at the protein level and subsequently produces additional cAMP to initiate the downstream signaling of Dop1[8,25,26]. Except for the transient activation on the catalytic activity, the increased *ac2* expression at the post transcriptional level via inhibition of miRNAs indicates a strengthened activation of Dop1 signaling. Thus, the double regulation of *ac2* at expression and activity in Dop1 signaling might provide an efficient and durable activation of the downstream cascade underlying the rapid behavioral responses.

Dop1 regulates miR-9a expression by inhibiting its maturation from pre-miR-9a, instead of regulating pri-miR-9a transcription and pre-miR-9a processing. MiRNA biogenesis is a complex process that involves numerous modulators, including Drosha, DGCR8, Dicer, and TRBP, and etc.[43]. The regulation of miR-9a by Dop1 at the later step in miR-9a biogenesis may reduce the effect of other factors on its biogenesis at the earlier steps. The regulation efficiency is supposedly increased by the specific regulation of miR-9a maturation as compared with the other steps. Previous studies showed that membrane receptor signaling, including epidermal growth factor receptor and transforming growth factor beta (TGF-beta)-SMAD signaling, modulate miRNA processing and maturation through the DROSHA and AGO2 complexes, respectively[50,51]. In the present study, we revealed a new mechanism by which Dop1, a typical GPCR, regulates the maturation of miR-9a through the La protein.

In this study, Dop1-regulated miR-9a maturation by inhibiting the binding of the La protein to pre-miR-9a. As a highly abundant phosphoprotein, La can recognize the stem-loop structure of

 **11**

pre-miRNA and is required for miRNA expression[44]. Metabolic labeling experiments revealed that La proteins from humans to yeast are phosphorylated in vivo[52,53]. The phosphorylation of the La protein by Akt significantly impairs its RNA chaperone activity and contributes to the specific mRNA translation in glial progenitors[54]. Moreover, activation of D1 receptors phosphorylates the downstream proteins through the canonical AC–cAMP–PKA pathway and the non-canonical Akt and ERK pathways[8]. Thus, Dop1 activation might increase the phosphorylation of La protein, impair its binding efficiency to pre-miR-9a, and trigger its disassociation from the stem–loop structure of pre-miR-9a.

La protein has been proved to promote miRNA biogenesis by stabilizing pre-miRNAs from nuclease-mediated decay in HeLa cells[44]. However, in the migratory locusts, the expression level of pre-miR-9a did not change after La RNAi knockdown or Dop1 activation. These results suggested that La protein may not maintain the level of pre-miR-9a through preventing nuclease-mediated decay. The results in human breast cancer cells suggested that cooperative relationships between La/SSB and Dicer allowed pre-miRNAs to be easily transferred from La/SSB to the Dicer complex for maturation processing[44]. Thus, La protein possibly facilitates the biogenesis of miR-9a from pre-miR-9a with other cooperative regulators, such as Dicer. The molecular mechanisms by which La protein facilitates miR-9a maturation need to be further investigated.

In summary, our results reveal a Dop1–miR-9a–AC2 circuit underlying the olfactory responses in the migratory locust. This circuit suggests a novel molecular mechanism by which Dop1 mediates behavioral plasticity. Furthermore, the regulation of miRNA biogenesis by the GPCR receptor at the post transcriptional level could increase our understanding of GPCR function in mediating the interaction between genes and environments.

## Methods

**Animal husbandry**. Experiments were performed using fourth-stadium migratory locusts (*L. migratoria*) from the solitarious and gregarious colonies maintained in the Institute of Zoology, Chinese Academy of Sciences, Beijing, China. Gregarious locusts were cultured in large boxes (40×40×40 cm) at a density of 500–1000 locusts per container. Solitarious locusts were obtained from the gregarious colony and cultured individually in separate white metal boxes (10×10×25 cm). These boxes were supplied with charcoal-filtered compressed air. The gregarious and solitarious locusts were maintained for at least three generations before the experiments were conducted. The gregarious and solitarious colonies were maintained in a 14 h light/10 h dark cycle at 30 ± 2 °C and fed with fresh wheat seedlings and bran[55].

**Olfactory behavior assay in Y-maze**. The olfactory behavior was examined in a Y-maze. The diagram of the Y-maze (Supplementary Figure 1) was placed in a room specific for olfactory behavioral assay. The Y-tube was placed in an enclosed and sound insulated cage with a transparent window, to avoid the disturbance from the observer or other ones during behavioral assay. Each locust was observed for 4 min and examined only once. Whenever the individual locust moved more than 4 cm into the volatile or air arm within 4 min, this individual locust was recorded and considered as the first choice for the respective arm.

The odorants used were the volatiles emitted from the body and feces of 40 fourth-stadium gregarious locusts. The components and corresponding concentrations were analyzed and listed in the previous study[24]. The volatiles were delivered to either arm to eliminate possible spatial bias. The air flow was set at 300 mLmin$^{-1}$.

We assessed olfactory preferences by giving each locust a single choice between gregarious volatiles and air in a Y-maze, and each locust is a replicate. For statistical analysis, we pooled all insects tested and calculated the significance using *G*-test for independence. To quantify the choice behavior, we directly calculated the percentage of locusts that selected the volatile arm to all tested locusts.

To assess the specificity of attraction in migratory locusts to the gregarious volatiles, the volatiles from other species (*Solenopsis invicta*, *Acheta domestica*, *Blattella germanica*, *Apis mellifera*, and *Helicoverpa armigera*) were presented in one arm of a Y-maze. The total body weight of other species was the same as that of 40 fourth-stadium gregarious locusts to eliminate the individual differences among species. The behavioral assay was performed as described above.

**Protein preparation and Western blot analysis**. Six samples of locust brains (8–10 individuals/sample) were collected and homogenized in Trizol reagent (Life Technology) and protein for Western blot analysis was extracted following manufacturers' instructions. For protein analyses, we developed affinity-purified polyclonal antibodies against Dop1 (rabbit) and AC2 (mouse), and a monoclonal antibody against La protein (mouse) (Beijing Protein Innovation Co., Ltd., BPI) and the specificities of these antibodies were evaluated (Supplementary Figure 3B, 10, and 12B). We separated the protein samples (100 μg) using gel electrophoresis and transferred them onto polyvinylidene difluoride membranes (Millipore). Membranes were blocked for non-specific binding using 5% bovine serum albumin, then incubated with the primary antibodies (rabbit anti-Dop1 serum, 1:500; mouse anti-AC2 serum, 1:500; mouse anti-La serum, 1:1000) in TBS-T overnight at 4 °C. After incubation, the membranes were washed, incubated with anti-rabbit IgG or anti-mouse IgG secondary antibody (1:5000) (CWBIO, China) for 1 h at room temperature, and then washed again. 5-bromo-4-chloro-3-indolyl phosphate/nitroblue tetrazolium substrate (CWBIO, China) was then used for colorimetric detection. The antibodies were stripped from the blots, re-blocked, and then probed with an anti-GAPDH antibody (CWBIO, China). The intensities of the Western blot signals were quantified using densitometry. Images have been cropped for presentation in the Figs. 1c, 5d, f, 6c, d and 8b, and Supplementary Figures 3B, 10, and 12B. The original full size images are presented in Supplementary Figures 14, 15, and 16.

**Behavioral pharmacology**. The behavioral change from solitarious to gregarious phase is much slower than the change from gregarious to solitarious phase[23]. Dopamine fluctuation and receptor expression significantly changed in the brains of solitarious locusts after crowding for 4 h, and changed in the brains of the gregarious locusts after 1 h of isolation[19]. In the present study, the duration of pharmacology intervention is determined as 1 h in the gregarious locusts and 4 h in the solitarious locusts according to the previous study[19].

To determine the role of Dop1 in locust olfactory preference, we injected SKF38393 (Sigma–Aldrich) (5 mM × 69 nL) into the brains of fourth-stadium solitarious locusts and then assayed their olfactory behavior after 4 h. We injected SCH23390 (Sigma–Aldrich) (5 mM × 69 nL) into the gregarious locusts as described above and then assayed their olfactory behavior after 1 h. The control group received the same volume of saline before the behavioral assay.

To determine the role of AC2 in locust olfactory preference, we injected forskolin (Sigma–Aldrich) (5 mM × 69 nL) into the brains of fourth-stadium solitarious locusts and then assayed their olfactory behavior after 4 h. We injected SKF83566 (TOCRIS) (5 mM × 69 nL) into the gregarious locusts as described above and then assayed their olfactory behavior after 1 h. The control group received the same volume of saline before the behavioral assay.

**RNAi and behavior assay**. After designing the fragments of *Dop1*, *phf20l1*, *ac2*, and *La* sequences for RNAi, we blasted the target fragments against the genome sequences of the migratory locust[37] to detect sequence homologies. We selected the fragment that was non-homologous with other genes in the genome database to avoid non-specificity during RNAi knockdown. Double-stranded RNAs (dsRNAs) of green fluorescent protein (GFP), *Dop1*, *Phf20l1*, *Ac2*, and *La* were prepared using the T7 RiboMAX Express RNAi System (Promega, Madison, USA). Before injection, the fourth-stadium locusts were placed in a Kopf stereotaxic frame specially adapted for locust surgery. We used a pair of Nevis scissors to cut a midline incision in the central position between the paired compound eyes and expose the brains for injections. All injections were performed under a dissecting microscope using a NANOLITER injector 2000 (World Precision Instruments, Sarasota, FL, USA) with a glass micropipette tip. We injected 72 ng of dsDop1, dsAc2, and dsLa into the brains of fourth-stadium gregarious locusts and 72 ng of dsPhf20l1 into the brains of solitarious locusts. The effect of RNAi on relative mRNA and protein expression levels were investigated by qPCR and Western blot after injection for 72 h (Supplementary Figures 3, 7, 8, 10 and 12). The dsRNA-injected locusts were reared for 72 h before the behavioral assay to determine the role of these genes in locust olfactory responses. The duration for RNAi is determined according to the previous studies[19,20,23]. The primers for RNAi are provided in Supplementary Table 3. The nucleotide sequences for RNAi are listed in Supplementary Table 4.

**RNA preparation and qPCR assay for genes and miRNAs**. Eight samples of locust brains (8–10 individuals/sample) were collected and homogenized in Trizol reagent (Life Technology) and the total RNA was extracted following manufacturers' instructions. DNase was applied to eliminate DNA contamination in the RNA samples. We reverse transcribed 2 μg of total RNA in every sample using a miRcute miRNA First-Strand cDNA Synthesis Kit (Tiangen, Beijing, China) and M–MLV reverse transcriptase (Promega, Madison, USA) in accordance with the manufacturers' instructions to analyze the expression levels of miRNAs and genes, respectively. PCR amplification was conducted with the Roche Light Cycler 480 using a miRcute miRNA qPCR Detection Kit (Tiangen) and a Real Master-Mix (SYBR Green) kit (Tiangen, Beijing, China), respectively. U6 snRNA and RP49 were used as endogenous controls for miRNAs and mRNAs, respectively. The amplification procedure was following the manufacturer protocol of Kits and the melting curve was detected to confirm the amplification specificity of the target

genes. All PCR amplifications are sequenced to verify the specificity of primers. The primers and miRNA sequences for qPCR assay are provided in Supplementary Table 5 and Supplementary Table 6.

To selectively quantify pri-miR-9a, pre-miR-9a, and miR-9a, we designed the specific primers as presented in Supplementary Figure 13. We quantified pri-miR-9a in the cDNA samples reverse transcribed using M–MLV reverse transcriptase. The forward and reverse primers are both specific for pri-miR-9a and designed in the sequence without pre-miR-9a. We quantified pre-miR-9a and miR-9a in the cDNA samples reverse transcribed using a miRcute miRNA First-Strand cDNA Synthesis Kit (for short RNAs) (Tiangen). This kit adds Adaptor (specific sequence)-Poly(A) to the 3′-terminal of miRNA and synthesize the first-strand cDNA based on Poly(A) modified miRNA through oligo(dT)-universal tag primed reverse transcription. We respectively quantified pre-miR-9a and miR-9a using their specific forward primer and a reverse primer complementary to the adaptor provided in the qPCR kit. The forward primer of pre-miR-9a is complementary to the loop, which is absent in the miR-9a (Supplementary Figure 13). All PCR amplifications are sequenced to verify the specificity of primers for pri-miR-9a, pre-miR-9a, and miR-9a. The PCR amplifications of miR-9a are sequenced to verify pre-miR-9a are rarely amplified in these samples.

**Deep sequencing for small RNAs.** The brains of fourth-instar gregarious locusts were collected 72 h after injection of dsGFP or dsDop1 (72 ng). Similarly, the brains of fourth-instar solitarious locusts were collected 4 h after injection of saline or SKF38393 (5 mM). Each sample contained 30 brains. The concentration and purity of total RNA were measured in an Agilent 2100 Bioanalyzer (Agilent) to verify RNA integrity. Small RNA libraries were constructed using a TruSeq small RNA sample preparation kit (Illumina) in accordance with the manufacturer protocol. The protocol was designed on the basis of miRNA structure properties, i.e., most mature miRNAs have 5′-phosphate and 3′-hydroxyl groups. The RNA 3′ and RNA 5′ RNA adapters are ligated to their corresponding ends of RNA molecules. Afterward, the ligated RNA fragments were reverse transcribed using M-MLV reverse transcriptase (Life Technology). The resulting cDNAs were amplified with two primers that were complementary to the ends of the adapter sequences. After PCR amplification, the samples were separated according to size in a 6% Novex polyacrylamide gel to be enriched for miRNA molecules and were sequenced on an Illumina Genome Analyzer IIx sequencing system. The raw sequencing data are available for download form the NCBI SRA server (accession number: SRP116154).

Unsupervised hierarchical clustering was performed using Cluster 3.0[56], which employs uncentered Pearson correlation and average linkage; the results are presented by Java Treeview software[57]. We determined the clusters a, b, c, and d according to the expression patterns of the miRNAs in SKF and dsDop1 groups. Cluster a includes the miRNAs, which are downregulated in SKF group and up-regulated in dsDop1 group. Cluster b includes the miRNAs downregulated in both SKF and dsDop1 groups. Cluster c includes the miRNAs upregulated in both SKF and dsDop1 groups. Cluster d includes the miRNAs up-regulated in SKF group and downregulated in dsDop1 group. The thermal graph includes the miRNAs showed significant change in either group or both.

**Overexpression of miR-9a in S2 cells.** To overexpress miR-9a in *Drosophila* S2 cells, we constructed a miRNA-expressing vector using the pAc5.1/V5-His B vector (Life Technology). About 500 bp fragments centered around the pre-miR-9a or pre-miR-315-5p (negative control) sequence were amplified from locust genome DNA with Kpn I and Xho I sites using HS prime star DNA Polymerase (Takara). Double enzyme-digested PCR products can be ligated into the pAc5.1/V5-His B vector, which was previously digested with Kpn I and Xho I. The efficiency of overexpression was verified by qPCR, and the primers are listed in Supplementary Table 7.

**In vitro luciferase reporter gene assays.** The 200–500 bp sequences of CDS or 3′UTR surrounding the predicted miR-9a target sites in *gart*, *amd1*, *sirt3*, *phf20l1*, and *ac2* were separately cloned into the psiCHECK-2 vector (Promega, Madison, USA) using the Xho I and Not I sites. Mutagenesis PCR was performed at the miR-9a target sites by using the Fast Mutagenesis System (TransGen). The point mutations in the putative target sites (WT, orange) were engineered in the region complementary to the miR-9a seed sequence (red). Those point mutations of MT sequences are indicated in blue. About 10 h before transfection, 0.5 mL of $1 \times 10^6$ cells mL$^{-1}$ S2 cells were seeded into each well of a 24-well plate with complete Schneider's *Drosophila* medium. S2 cells in a 24-well plate were co-transfected with 20 ng of the luciferase reporter vector (WT or MT) and 100 ng of the miR-9a expression plasmid using cellfectin II (Life Technology). The activities of the firefly and Renilla luciferases were measured 48 h after transfection with the Dual-Glo Luciferase Assay System (Promega, Madison, USA) using a luminometer (Promega, Madison, USA).

**Target prediction of miR-9a using miRanda and RNAhybrid.** We employed the algorithms miRanda and RNAhybrid to predict the potential target genes of miR-9a in the gene database from locust transcriptome. The alignment score of miR-anda was restricted to 140, and other parameters are default. The parameters of RNAhybrid is set as 1 hits per target, free energy threshold is −17 kcal mol$^{-1}$, helix

constraint is 2 to 6, max bulge loop length is 2, max internal loop length is 6, no G:U in seed is included. The predicted results are listed in Supplementary Data 1, which include the accession number of genes in the transcriptome, gene lengths, position of alignment, predict score, gene names, and GO categories.

**RNA-seq and data processing.** The brains of forth-instar gregarious and soli-tarious locusts were collected, and each sample contained 10 brains (5 males and 5 females). Three independent replicates were performed for each group. Total RNA was isolated as previously described, and RNA quality was confirmed by agarose gel. cDNA libraries were prepared according to Illumina's protocols. After paired-end sequencing, the raw reads were assembled by Trinity software (version 2011-08-20) to obtain reference transcripts due to locust genome data. Downstream analysis of alignment (Bowtie) and abundance estimation (RSEM) were performed using the utility package in Trinity software (version 2011-08-20). The differentially expressed genes (DEGs) were analyzed using EdgeR software. The DEGs with fold change > 1.5 and $P < 0.05$ were selected. The accession number of genes in the transcriptome, gene length, Log2(G/S), $P$ value, position of alignment, and annotation were listed in Supplementary Data 2. Finally, Blast2Go[58] was used to annotate and enrich the DEGs. The raw reads of 6 samples are available for download form the NCBI SRA server (accession number: SRP116073).

**miRNA agomir and antagomir treatment in vivo.** In vivo delivery of the miRNA agomir–a chemically modified, cholesterylated, stable miRNA mimic—results in target silencing, similar to silencing using overexpression of endogenous miRNA[21]. Thus, "miR-9a overexpression" in this study represents the expression of activated agomir-9a. The miRNA antagomir is a chemically modified, cholesterol conjugated, single-stranded RNA analog that complements the miRNAs; it efficiently and specifically silences endogenous miRNAs[21]. The brains of 2-day-old gregarious or solitarious fourth-instar nymphs were microinjected with agomir-9a or antag-omir-9a, respectively. The sequence of a *Caenorhabditis elegans* miRNA, cel-miR-67-3p (5′-UCA CAA CCU CCU AGA AAG AGU AGA-3′), was used as a negative control. In brief, we placed fourth-instar nymphs in a Kopf stereotaxic frame that was adapted for locust surgery. Using Nevis scissors, we made midline incision of about 2 mm in length at the mid-point between the two antennae, exposing the underlying brain. Following this, 42 pmol of agomir-9a or antagomir-9a (200 mM; RiboBio) was injected into the brains. The agomir-negative or antagomir-negative controls (200 mM) were also injected into the gregarious or solitarious locust brains (RiboBio, China). All injections were administered using a nanoliter injector (World Precision Instruments) with a glass micropipette tip under an anatomical lens. The duration of agomir and antagomir treatment is determined according to the previous study[21]. The treated nymphs were subjected to behavioral analysis after 48 h. Their brains were harvested, snap-frozen, and then stored at −80 °C.

**Behavioral rescue experiments in vivo.** To determine the role of miR-9a in locust olfactory attraction, we injected 42 pmol of agomir-9a into the solitarious brains. After 48 h before behavioral assay, SKF38393 (5 mM × 69 nL) was injected into the brains of solitarious locusts. The injected locusts were then raised under the soli-tarious condition and subjected to behavioral analysis 4 h after injection of SKF38393. Control insects were only treated with an equal amount of SKF38393 or saline.

To determine the role of AC2 in locust olfactory attraction, we injected 72 ng of dsAc2 into the solitarious brains. After 72 h before behavioral assay, SKF38393 (5 mM × 69 nL) was injected into the brains of solitarious locusts. The injected locusts were then raised under the solitarious condition and subjected to behavioral analysis 4 h after injection of SKF38393. Control insects were only treated with an equal amount of SKF38393 or saline.

**Fluorescence in situ hybridization of miR-9a and ac2.** The in situ analysis of miR-9a and *ac2* in the locust brain was determined by the co-labeling of miRNA and mRNA FISH. An antisense locked nucleic acid detection probe for miR-9a or a negative control was labeled with double digoxigenin (Exiqon). The *ac2* fragment (861 bp) was prepared for antisense and sense probes. We blasted these two fragments against the genome sequences of the migratory locust to detect homo-logies and to avoid non-specificity in hybridization. The sequence of primers for *ac2* fragment were forward, 5′- GGA TGG CAC ATC ACA GAA AC -3′; reverse, 5′- TAT TTA CAG CAT CAC CCC AG -3′. The brains were fixed in 4% paraf-ormaldehyde overnight at 4 °C. Thick sections of the brains (50 μm) were cut with a vibrating blade microtome (VT 1200 S, Leica, Wetzlar, Germany) and washed with 0.1 M PBS for 15 min. The brain tissue slides were digested with 20 mg mL$^{-1}$ proteinase K (Roche) at 37 °C for 15 min, washed three times with 0.1 M PBS for 15 min each (pH 7.4), and then re-fixed before placing in the pre-hybridization solution (Boster, Wuhan, China) for 2 h at 37 °C. We used pre-hybridization solution containing biotin-labeled miR-9a probe (2 nM) and digoxigenin-labeled *ac2* probe (3 μg mL$^{-1}$) for hybridization overnight in a humidified chamber at 37 °C. Washing was conducted in 4×, 2×, 1×, and 0.2× SSC at 37 °C for 30 min each. Double FISH with Dig- and biotin-labeled probes was visualized by the anti-Dig AP-conjugated antibody in combination with HNPP/Fast Red (Roche). The TSA kit (Perkin Elmer, MA, USA), including an anti-biotin strepavidin horseradish peroxidase conjugate and fluorescein-tyramides as substrate, was used for the

biotin-labeled probes. The sequence of a *Caenorhabditis elegans* miRNA, cel-miR-67-3p (5′-UCA CAA CCU CCU AGA AAG AGU AGA-3′) and sense sequence of *ac2* probes were used as the negative control. The nucleus of locust brain is labeled by Hoechst33342 (Life Technology) to indicate the brain structure. miR-9a and *ac2* signals were detected under a LSM 710 confocal fluorescence microscope (Zeiss Oberkochen, Germany).

**Immunofluorescence of Dop1 and AC2 in the locust brain**. After the dissected brain tissues were fixed in 4% formaldehyde at 4 °C overnight, they were washed with 0.1 M PBS twice for 15 min each (pH 7.4). Thick sections of the brains (30 μm) were cut with a vibrating blade microtome (VT 1200 S, Leica, Wetzlar, Germany), washed in 0.1 M PBS for 15 min, and then incubated in 0.1 M PBS containing 5% normal goat serum (NGS, Boster, China) for 1 h at room temperature. The primary anti-Dop1 and anti-AC2 (custom made, see "Protein preparation and Western blot analysis" section of Methods for details) was diluted at 1:300 in 0.1 M PBS containing 2% NGS. Incubation with primary antibodies lasted for 48 h. The tissues were washed with 0.1 M PBS three times for 15 min each and subsequently incubated with the mixture of two secondary antibodies, Goat anti-rabbit antibody Alexa fluor 488 (1: 500, A11034, Life Technology) and Goat anti-mouse antibody Alexa fluor 546 (1: 500, A11030, Life Technology), for 1 h at room temperature. After washing three times, the tissues were mounted in anti-fade fluorescence mounting medium. The negative serum of Dop1 and AC2 from rabbit and mouse were used as the negative control. The nucleus of locust brain is labeled by Hoechst33342 (Life Technology) to indicate the brain structure. The fluorescence was detected using a Zeiss LSM 710 confocal microscope (Zeiss, Oberkochen, Germany).

**Total RNA extraction from nuclear and cytoplasmic fractions**. The separation of nuclear and cytoplasmic fractions are referred to the previous study[32] with slight modifications. The brains were homogenized in cold PBS containing 0.2% Nonidet P-40. The lysate was centrifuged at $30 \times g$ for 2 min at 4 °C to remove the insoluble fragment of tissue. Then, the supernatant was centrifuged at $1500 \times g$ for 15 min at 4 °C. The nuclei were in the pellet whereas the cytoplasm remained in the supernatant. The nuclei were resuspended two times in PBS to remove the residual cytoplasm. The cytoplasmic fraction was centrifuged at $2000 \times g$ for 10 min at 4 °C to remove the residual nuclei. The nuclei and cytoplasm were added 1 mL Trizol and the total RNA in nuclear and cytoplasmic fractions are extracted as described above. U6 was used as a nuclear marker and 18 S rRNA as a cytoplasmic marker to verify the fraction quality.

**RNA immunoprecipitation assay**. The protocol for the RIP assay for *ac2* was modified from the Magna RIP Quad kit (Millipore). The monoclonal antibody against Ago-1 protein was developed in mice (Abmart)[21]. About 40 brains were collected and homogenized in ice-cold RIP lysis buffer, then stored at −80 °C overnight. Magnetic beads were pre-incubated with 5 μg of Ago-1 antibody (Abmart) or with 5 μg of normal mouse IgG (Millipore), which was used as a negative control. Next, the frozen homogenates were thawed and centrifuged, and the supernatant was incubated overnight with the magnetic bead–antibody complex at 4 °C. Meanwhile, 1/3 of the lysate was stored as "input" sample. The RNA in the immunoprecipitates and inputs was extracted by Trizol reagent (Life Technology) and was purified and reverse-transcribed into cDNA using the high-capacity RNA-to-cDNA Kit (ABI). RT-PCR and qPCR were performed using these cDNAs as templates to quantify the *ac2* mRNA. To normalize the relative expression levels, the supernatants of the RIP lysate (input) and the IgG controls were assayed for specificity of RNA–protein interactions.

The RIP assay for pre-miR-9a was performed as above. The monoclonal antibody against La protein, which was developed in mice (BPI) (Supplementary Figure 12B), was used to precipitate pre-miR-9a in the locust brains. Magnetic beads were pre-incubated with 5 μg of La protein or with 5 μg of normal mouse IgG (Millipore). The RNA in the immunoprecipitates and inputs was extracted by Trizol reagent (Life Technology) and was purified and reverse-transcribed into cDNA using a miRcute miRNA First-Strand cDNA Synthesis Kit (Tiangen, Beijing, China). RT-PCR and qPCR were performed using these cDNAs as templates to quantify the pre-miR-9a expression. The supernatants of the RIP lysate (input) and the IgG controls were assayed to normalize the relative expression levels and ensure the specificity of the RNA–protein interactions.

**RNase-assisted RNA pull-down and mass spectrometry**. RNA pull-down and mass spectrometry were performed as previously described[59] with some modifications. In brief, synthetic pre-miR-9a with biotin-labels at the 3′-terminal were chemically coupled to streptavidin magnetic beads (from Pierce™ Magnetic RNA-Protein Pull-Down Kit, 20164, Thermo Fisher). The streptavidin magnetic beads (50 μL) were washed for 3 times using 20 mM Tris (pH 7.5) and were incubated with biotin-labeled pre-miR-9a (50 pmol) in 1 × RNA capture buffer (20 mM Tris (pH 7.5, 1 M NaCl, 1 mM EDTA) at room temperature for 30 min. The synthetic RNAs used are listed in Supplementary Table 1. The total protein extracts from brains injected with saline, SKF38393, dsGFP, and dsDop1 were incubated with

magnetic beads with pre-miR-9a at 4 °C for 60 min. The detailed operation was following manufacturers' instructions of Pierce™ Magnetic RNA-Protein Pull-Down Kit. The incubation was followed by a series of washes with Roeder D buffer (100 mM KCl, 20% (v/v) glycerol, 0.2 mM EDTA, 100 mM Tris pH 8.0). After the final washing, the beads with associated proteins were re-suspended in structure buffer (10 mM Tris-HCl pH 7.2, 1 mM MgCl₂, 40 mM NaCl) and then treated with RNases A/T1 (Fermentas) at 37 °C for 30 min. The samples were subsequently analyzed by SDS-PAGE, followed by mass spectrometry (BPI) or Western blot.

The bands in SDS-PAGE gels were accurately cut and collected in an Eppendorf tube for mass spectrometry analysis in BPI. The proteins were digested to peptides and centrifuged at $20,000 \times g$ for 15 min. After that, 10 μL supernatant was loaded on a ultimate3000 nanoLC (Dionex) by the autosampler onto a 2 cm C18 trap column. Then, the peptides were eluted onto a 10 cm analytical C18 column (inner diameter 75 μm) packed in-house. The separated peptides were subjected to nanoelectrospray ionization followed by tandem mass spectrometry (MS/MS) in a Q Exactive (Thermo scientific) coupled online to the HPLC. Peptides were selected for MS/MS using (HCD) operating mode with a normalized collision energy setting of 28%. Intact peptides were detected in the Orbitrap at a resolution of 70,000 (first order) and 17,500 (second order). The score for each peptide indicates ions score that is $-10 \log(P)$, where $P$ is the probability that the observed match is a random event. In the present study, individual ions score >36 indicates identity or extensive homology ($P < 0.05$), and these peptides are potential candidates to be further tested using Western blot.

**GST–La protein pull-down assay**. Full-length GST–La fusion proteins and GST proteins were expressed in *Escherichia coli* and then bound to glutathione-agarose beads (Life Technology). GST–La and GST bound beads were washed, re-suspended in bind buffer (20 mM Tris-HCl, pH 7.6, 100 mM KCl, 0.1% Tween-20, and 0.1% Triton X-100), incubated with the chemically synthetic RNA pre-miR-9a (50 pmol) for 1 h at 4 °C, and then washed four times with binding buffer. The bound RNA was eluted by adding the elution buffer (1% SDS and 150 mM NaCl) for 15 min at room temperature. Trizol reagent (Life technology) was added, and RNA was purified using the usual method. RNA was resuspended in water and used in a reverse transcription reaction, followed by RT-PCR and qPCR to detect the relative levels of RNA binding to GST–La fusion proteins. Four biological replicates were used in both GST–La and GST groups.

**Statistical analysis**. The differences between control and treatment group in qPCR and Western blot were compared using Student's t-test. The statistical analysis of olfactory preference was referred to the work on locust aversive learning[60] and olfactory attraction/repulsion[22]. In the statistical analysis of olfactory preference, G-test for goodness-of-fit was used to determine the significance of the divergence from an expected 50% decision for the volatile arm or air control arm, and the counts of migratory locusts preferring the volatile or air control were used for this analysis. The comparison between gregarious volatiles and other insect volatiles is the same. The olfactory responses between the control and treatment were analyzed using G-test for independence, and quantities of all locusts tested were included for this analysis. The data of behavioral assay is presented as $p \pm$ SE. The standard error (SE) of locusts' volatile preference was calculated as $\sqrt{p(1-p)/n}$, where $p$ is the proportion of locusts that were attracted by the volatiles and $n$ is the number of locusts that were tested. Differences were considered significant at $P < 0.05$. All statistics were analyzed using SPSS 17.0 (SPSS Inc., Chicago, IL, USA).

**Data availability**. Sequence data of *Dop1*, *ac2* and *La* that support the findings of this study have been deposited in Genbank with the accession codes (KP780182), (MG917090), and (MG917091), respectively. All data supporting the findings of this study are available within the article and its Supplementary Information files.

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

## Acknowledgements

We thank Prof. Zhongsheng Sun for his expert technical assistance on this study. We thank Dr. Li Chen for providing fire ants in behavioral assay. We thank Dr. Yundan

Wang for providing GAPDH antibody in Western blot. This research was supported by Strategic Priority Research Program of the Chinese Academy of Sciences (no. XDB11010200) and the National Natural Science Foundation of China (grant nos. 31430023, 31600850, 31522054, 31472040, and 31210103915).

## Author contributions

The author(s) have made the following declarations about their contributions. X.G, Z.M., and L.K. conceived and designed the experiments. X.G., T.L, W.L., and L.X. performed the experiments. X.G., B.D., J.H., and L.K. analyzed the data. X.G., Z.M., and L.K. wrote the manuscript.

## Additional information

**Competing interests:** The authors declare no competing interests.

