## [Peer Review File(PDF 1177 kb) · Nature Communications]

Reviewers' comments:

Reviewer #1 (Remarks to the Author):

In this manuscript, Guo et al. showed the molecular mechanism of dopamine receptor signaling underlying olfactory attraction that is involved in locust aggregation in migratory locust. They found that the specific miRNA (miR-9a) is a critical regulator of dop1-mediated olfactory attraction in gregarious locusts. They also identified an adenylyl cyclase (AC2) as a major target of miR-9a. Overall, the work is thorough, and evidence seems convincing. Following points should be addressed before publication.

1. The same authors have already reported that miRNA and dopamine are involved in aggregation behavior in locust (Yang M et al., 2014). It is critical to make clear what is novel over this (and related) reports.
2. The authors should provide how they computed the results in the thermal graph in Figure 2B. The results in Fig. 2B do not seem to represent the results in Fig. 2C and 2D (compare the values for miR-9a, for example). More explanation is needed.
3. The authors should provide technical details regarding how the authors could selectively quantify mature miRNA without detecting primary- or pre-miRNA.
4. In Figure 2B, locusts used are not annotated as solitary or gregarious, also it is not explained in the text.
5. The overall writing requires more rigor. Just to give some examples:
 - 1) The legends of Figure 3 and Figure 4 seem mismatched.
 - 2) Annotation of significance ("a" "b") in Figure 3D is lacking.
 - 3) Annotation of abbreviation "NC" (negative control) is spotted in the legend of Figure 5, but not in that of Figure 3.
 - 4) Colors of the letters in Materials and methods are not uniformed.
 - 5) Citations (especially in the Introduction) should be accurate. References contain mixed papers in mammals, locusts, culture cells, etc., and are sometimes not inappropriate.

Reviewer #2 (Remarks to the Author):

The present study demonstrates how neurotransmitter mediated control of microRNA maturation can influence rapid behavioral changes in *L. migratoria*. Specifically, the authors show that activated Dopamine Receptor 1 (Dop1) mediates olfaction-based locust gregariousness by inhibiting miR-9a maturation. Dop1 does so by inducing the dissociation of an RNA binding protein La from pre-miR-9a. This promotes the expression of miR-9a's target gene adenylyl cyclase 2 (ac2) in the higher olfactory processing centres. The claims of this study are very well supported by the data. The behavioral rescue experiments establish the validity of the posited Dop1-miR-9a-Ac2 circuit effectively. Further, RNA-IP assays provide sufficient evidence to establish Ac2 as a legitimate *in vivo* target of miR-9a in the brain. La binding to pre-miR-9a was also effectively demonstrated by pre-miR-9a RNA pull downs as well as La protein immunoprecipitation assays.

There are however a few issues which require clarification

Bioinformatics analysis to pull out miR-9a targets is somewhat opaque. Supplementary tables S1 and S2 are not clearly laid out. Table S2 is missing headers and the process of target gene selection is unclear since the source of transcriptome data and the cutoffs used for selecting target genes were not listed. Similarly Table S1 is hard to understand and would be served well by clear formatting and/or explanatory text.

In Fig S6 and S8 the nature of the negative control is not mentioned.

The assay times after agonist/antagonist treatment in Fig 1E and 1F are different - 1 hr or 4 hrs. I am assuming this is because G to S transition is fast and S to G is slow. Updating the methods to include the rationale for different testing times would be appropriate.

In Fig 4E (Fig 3 in Ms) the effect of Phf2011 knockdown on behavior was tested in S animals instead of G animals. If the rationale is that miR-9a targets Phf2011, then the behavior assay should be conducted in G animals to test loss of preference on target knockdown. I assume this was done because relative mRNA levels of Phf2011 were higher in S vs G brains. This overlooks the possibility of transient regulation of Phf2011 by miR-9a independent of other mechanisms which might serve to regulate Phf2011 in brains of long term S or G animals.

Minor details

Figure panels 3 and 4 have been interchanged.

In Fig 8A decrease in La band intensity on SFK treatment is difficult to visualize.

Primer sequence for Pri-miR-9a qPCR assays is not listed.

References to establish specificity of agonist/antagonists used are missing.

Line 167 typing error - dsGFP should be dsDop1.

Since, in mammals, bound La serves to protect pre-miRs from nucleases, it is interesting that there is no change in pre-miR-9a levels on La knockdown and/or Dop1 activation by SKF. This suggests that there must be an alternative mechanism by which La-binding increases mature microRNA levels.

Overall, this work presents an exciting mechanistic link between environment sensing, behavioral plasticity and microRNA regulation. I strongly recommend the acceptance of this study with the manuscript edited to address the highlighted concerns.

Reviewer #3 (Remarks to the Author):

The paper addresses interesting issues and reports interesting results. Unfortunately, the work is not well presented and there are several important aspects that need to be addressed before the work can be published.

- 1) The main behavioral assay, upon which the entire paper depends, is a Y-maze to test preference between no odor and volatiles from gregarious locusts. Unfortunately, no information is given on whether the observed differences really are due to differences in odor preference rather than differences in arousal/inquisitiveness.
- 2) More information should be given on the source of volatiles and their concentrations
- 3) Is the protocerebrum not more associated more with vision than olfaction?
- 4) There should be a detailed section on bioinformatic methods with information on stringency parameters, etc. (e.g. mirDeep, miRanda).
- 5) What is the reason for the differences in timing (48hr, 72 hr, 1hr, 4hr...). No explanation is provided.
- 6) Would be good to also show data points rather than bars +/- sem on the graphs
- 7) The behavioral experiments should be more clearly describe in the text. For example, it should be made clear that experiments are done with a Y-maze.
- 8) Samples sizes are not clear. It is mentioned that "Every experiment encompassed at least six replicates, and every replicate contained at least eight insects." But then for their statistics it is mentioned that $n > 40$. This raises the question of what are the "replicates" and whether there are issues of pseudoreplication. Also why not using fisher's exact instead of the G-test?
- 9) How many animals brains are used for analyses in Fig 1? Are these pooled samples?
- 10) How are clusters determined? Never mentioned.
- 11) The agomir-NC is not explained in the legend or text.
- 12) The mutant mir9a is not explained in the legend (what is mutated? we don't know.) or methods.
- 13) The entire bit on cytoplasm vs. nucleus doesn't seem useful. I don't see the value to the story. Also they never describe the methods for the fractionation.
- 14) The authors never describe the mass spec methods or explain their "identify score".
- 15) The ac2 western blot are a bit messy, which gives doubt to the quality of the densitometry quantification. All the more reason to see the points rather than a bar plot.
- 16) Seems that figures 3 and 4 are swapped.

In conclusion, the paper reports potentially interesting data, but it is difficult to evaluate the work given the lack of information. Editing would also make it easier to read.

Adria Leboeuf and Laurent Keller

Reviewers' comments:

Reviewer #1 (Remarks to the Author):

In this manuscript, Guo et al. showed the molecular mechanism of dopamine receptor signaling underlying olfactory attraction that is involved in locust aggregation in migratory locust. They found that the specific miRNA (miR-9a) is a critical regulator of dop1-mediated olfactory attraction in gregarious locusts. They also identified an adenylate cyclase (AC2) as a major target of miR-9a. Overall, the work is thorough, and evidence seems convincing. Following points should be addressed before publication.

- 1. The same authors have already reported that miRNA and dopamine are involved in aggregation behavior in locust (Yang M et al., 2014). It is critical to make clear what is novel over this (and related) reports.***

We thank the reviewer for his/her important comments. Dopamine can modulate animal behaviors at different levels, such as synthesis, release, reception and signaling, and neuron signal as we represented in introduction. The functions of dopamine in locust aggregation behavior are actually studied at different levels step by step in our group.

Dopamine pathway is conserved in animals (see Figure), in which many genes are involved in dopamine synthesis (e.g. *henna*, *pale*, *Ddc* etc.), transport and release (e.g. *vat1*) and reception (e.g. *Dop1*). Ma et al. (2011) have found that dopamine and related genes in its synthesis and release are involved in the locust aggregation behavior. Yang et al. (2014) have found that miR-133 inhibited dopamine synthesis through targeting two key genes (*henna* and *pale*) in this pathway. Furthermore, Guo et al. (2015) demonstrated that dopamine modulated locust aggregation through dopamine receptor 1 (Dop1), but not Dop2. The present study demonstrates that Dop1 inhibits miR-9a maturation to enhance olfactory attraction in aggregation in migratory locusts, thus this discovery focuses on the downstream signaling of the pathway. We further explore the novel mechanism that Dop1 mediates olfactory response through mediating maturation of miR-9a. The results in this study deepened and extended our understanding of the molecular mechanism of Dop1 signaling in locust aggregation

behavior.

To state clearly what is novel over these reports, we added related discussion in the Discussion section. We added “The function of dopamine in regulating aggregation behavior is closely correlated with epigenetic regulator miRNAs. After Ma et al. (2011) reported that dopamine and its related genes for synthesis (*pale* and *henna*), transport and release (*vat1*) were involved in the locust aggregation behavior. Yang et al. (2014) found that miR-133 inhibits dopamine synthesis underlying behavioral aggregation through inhibiting two key genes, *pale* and *henna*. However, it is unknown whether and how dopamine mediates the expression of the miRNAs through dopamine receptors with downstream signal transduction system. Followed by the finding of Dop1 mediating locust aggregation behavior¹⁹, we further found that Dop1 inhibits the maturation of miRNA-9a for olfactory behavioral regulation and the expression regulation of miR-133 is not correlated with Dop1 signaling. The novel mechanism that Dop1 modulates miR-9a maturation extends our understanding of dopamine action mode, and also provides a more comprehensive understanding of the function of dopamine in the migratory locust.” in Line 339-351.

2. *The authors should provide how they computed the results in the thermal graph in Figure 2B. The results in Fig. 2B do not seem to represent the results in Fig. 2C and 2D (compare the values for miR-9a, for example). More explanation is needed.*

We added more details on bioinformatic computing the results in thermal graph in Figure 2B in Materials and methods section. We added “Unsupervised hierarchical clustering was performed using Clustal 3.0⁵⁶, which employs uncentered Pearson correlation and average linkage; the results are presented by Java Treeview software⁵⁷. We determined the clusters a, b, c, and d according to the expression patterns of the miRNAs in SKF and dsDop1 groups. Cluster a includes the miRNAs, which are down-regulated in SKF group and up-regulated in dsDop1 group. Cluster b includes the miRNAs down-regulated in both SKF and dsDop1 groups. Cluster c includes the miRNAs up-regulated in both SKF and dsDop1 groups. Cluster d includes the miRNAs up-regulated in SKF group and down-regulated in dsDop1 group. The thermal graph includes the miRNAs showed significant change in either group or both.” in Line 566-575.

The mismatch between the results from RNA-seq and qPCR (genes and miRNAs) is actually in the previous studies of the migratory locust (Chen et al., 2010; Hou et al., 2015; Wei et al., 2009; He et al., 2016) and also in the present study. This may arise from the genome size of the migratory locusts (6.5 G), the largest genome in animals sequenced so far. The limited sequencing depth may generate the bias between RNA-seq results and qPCR. Thus, we conducted qPCR experiments, the more stable method for quantification, to confirm miRNA sequencing results and ensure the candidate miRNAs we chose from RNA-seq are closely related to the treatment.

In RNA-seq, miR-9a was significantly up-regulated in Dop1 group, but only showed a slight down-regulation (with no significance) in SKF group. However, qPCR showed that miR-9a expression significantly up-regulated in Dop1 group and down-regulated in SKF group, respectively. qPCR experiments were conducted with 8 biological replicates and each sample pooled 8-10 brains of the locusts. Thus, miR-9a showed significant down-regulation in SKF group when more biological replicates were added. For other miRNAs, some accidental enrichments in RNA-seq are dismissed in qPCR, or some candidates with no significance in RNA-seq are enriched in qPCR. We chose the candidate miRNAs in the present study are largely dependent on the confirmation of the qPCR results, which represent more the general pattern in each group. According to the qPCR results, miR-9a showed the significant change in both SKF and dsDop1 groups. Thus, we chose miR-9a as the target for further investigation.

3. *The authors should provide technical details regarding how the authors could selectively quantify mature miRNA without detecting primary- or pre-miRNA.*

Thanks for the comments. We provided more details regarding the selective quantification of pri-miR-9a, pre-miR-9a and miR-9a in the Materials and methods section.

We added “To selectively quantify pri-miR-9a, pre-miR-9a and miR-9a, we designed the specific primers as presented in Figure S12. We quantified pri-miR-9a in the cDNA samples reverse transcribed using MMLV reverse transcriptase. The forward and reverse primers are both specific for pri-miR-9a and designed in the sequence without pre-miR-9a. We quantified pre-miR-9a and miR-9a in the cDNA samples reverse transcribed using a miRcute miRNA First-Strand cDNA Synthesis Kit (for short RNAs) (Tiangen). This kit adds Adaptor (specific sequence)-Poly(A) to the 3'-terminal of miRNA and synthesize the first-strand cDNA based on Poly(A) modified miRNA through oligo(dT)-universal tag primed reverse transcription. We respectively quantified pre-miR-9a and miR-9a using their specific forward primer and a reverse primer complementary to the adaptor provided in the qPCR kit. The forward primer of pre-miR-9a is complementary to the loop, which is absent in the miR-9a (Figure S12). All PCR amplifications are sequenced to verify the specificity of primers for pri-miR-9a, pre-miR-9a, and miR-9a.” in Line 535-548.

4. *In Figure 2B, locusts used are not annotated as solitary or gregarious, also it is not explained in the text.*

We corrected the annotation “SKF/Saline” to “S-SKF/S-Saline”, and “dsDop1/dsGFP” to “G-dsDop1/G-dsGFP” in Figure 2B. We explained it as “Using small RNA transcriptome analysis, we analyzed the expression profile of miRNAs in the solitary brains after Dop1 activation, and in the gregarious brains after Dop1 inhibition to determine the potential miRNAs associated with Dop1 (Figure 2A).” in Line 146-149.

5. The overall writing requires more rigor. Just to give some examples:

1) The legends of Figure 3 and Figure 4 seem mismatched.

We corrected the legends of Figure 3 and Figure 4.

2) Annotation of significance ("a" "b") in Figure 6D is lacking.

We labeled the significance with "*" instead of "a and b".

3) Annotation of abbreviation "NC" (negative control) is spotted in the legend of Figure 5, but not in that of Figure 3.

We added the abbreviation "NC, negative control" in the legend of Figure 3.

4) Colors of the letters in Materials and methods are not uniformed.

We checked the Materials and methods sections and corrected them.

5) Citations (especially in the Introduction) should be accurate. References contain mixed papers in mammals, locusts, culture cells, etc., and are sometimes not inappropriate.

The suggestions are very important. We separated the findings from cells, tissues, insects, or mammals, and cited the references respectively. In order to be more clearly, we added more information and details in the cited references. The details are represented in Introduction and Discussion section.

Reviewer #2 (Remarks to the Author):

*The present study demonstrates how neurotransmitter mediated control of microRNA maturation can influence rapid behavioral changes in *L. migratoria*. Specifically, the authors show that activated Dopamine Receptor 1 (Dop1) mediates olfaction-based locust gregariousness by inhibiting miR-9a maturation. Dop1 does so by inducing the dissociation of an RNA binding protein La from pre-miR-9a. This promotes the expression of miR-9a's target gene adenylyl cyclase 2 (ac2) in the higher olfactory processing centres. The claims of this study are very well supported by the data. The behavioral rescue experiments establish the validity of the posited Dop1-miR-9a-Ac2 circuit effectively. Further, RNA-IP assays provide sufficient evidence to establish Ac2 as a legitimate in vivo target of miR-9a in the brain. La binding to pre-miR-9a was also effectively demonstrated by pre-miR-9a RNA pull downs as well as La protein immunoprecipitation assays.*

There are however a few issues which require clarification

Bioinformatics analysis to pull out miR-9a targets is somewhat opaque. Supplementary tables S1 and S2 are not clearly laid out. Table S2 is missing headers and the process of target gene selection is unclear since the source of transcriptome data and the cutoffs used for selecting target genes were not listed. Similarly Table S1 is hard to understand and would be served well by clear formatting and/or explanatory text.

We appreciate the reviewer for his comments, which are similar to the Reviewer #1. We corrected some errors and added more details on the bioinformatics analysis in Materials and methods section, and revised the Table S1 and S2 to make it clearer.

We added “Unsupervised hierarchical clustering was performed using Clustal 3.0⁵⁶, which employs uncentered Pearson correlation and average linkage; the results are presented by Java Treeview software⁵⁷. We determined the clusters a, b, c, and d according to the expression patterns of the miRNAs in SKF and dsDop1 groups. Cluster a includes the miRNAs, which are down-regulated in SKF group and up-regulated in dsDop1 group. Cluster b includes the miRNAs down-regulated in both SKF and dsDop1 groups. Cluster c includes the miRNAs up-regulated in both SKF and dsDop1 groups. Cluster d includes the miRNAs up-regulated in SKF group and down-regulated in dsDop1 group. The thermal graph includes the miRNAs showed significant change in either group or both.” in Line 566-575.

We added “We employed the algorithms miRanda and RNAhybrid to predicate the potential target genes of miR-9a in the gene database from locust transcriptome. The alignment score of miRanda was restricted to 140, and other parameters are default. The parameters of RNAhybrid is set as 1 hits per target, free energy threshold is -17 kcal/mol, helix constraint is 2 to 6, max bulge loop length is 2, max internal loop length is 6, no G:U in seed is included. The predicted results are listed in Table

S1, which include the accession number of genes in the transcriptome, gene lengths, position of alignment.” in Line 601-607.

We added “The differentially expressed genes (DEGs) were analyzed using EdgeR software. The DEGs with fold change > 1.5 and $P < 0.05$ were selected. The accession number of genes in the transcriptome, gene length, $\text{Log}_2(\text{G/S})$, P value, position of alignment, and annotation were listed in Table S2. Finally, Blast2Go⁵⁸ was used to annotate and enrich the DEGs.” in Line 617-621.

We refined the information in Table S1 to make it clearer, and the details are presented in Materials and methods section in Line 601-607. We added headers and refined the information in Table S2. The details are presented in Line 617-621.

In Fig S6 and S8 the nature of the negative control is not mentioned.

Thanks for the comments. We added the information of Fig S6 and S8 in the text and also more details in Materials and methods section.

Figures S6 and S8 were changed into Figure S8 and S10 in the revised version. We added “The negative control showed no signal in the same part of the brain (Figure S8).” in Line 244-245, and “The negative control did not show any signal in the same part of the brain (Figure S10).” in Line 275-276, and “The sequence of a *Caenorhabditis elegans* miRNA, cel-miR-67-3p (5'-UCA CAA CCU CCU AGA AAG AGU AGA-3') and sense sequence of *ac2* probes were used as the negative control. The nucleus of locust brain is labeled by Hoechst33342 (Life Technology) to indicate the brain structure.” in Line 678-681.

We added “The negative serum of Dop1 and AC2 from rabbit and mouse were used as the negative control. The nucleus of locust brain is labeled by Hoechst33342 (Life Technology) to indicate the brain structure.” in Line 695-698.

The assay times after agonist/antagonist treatment in Fig 1E and 1F are different - 1 hr or 4 hrs. I am assuming this is because G to S transition is fast and S to G is slow. Updating the methods to include the rationale for different testing times would be appropriate.

Yes. We chose different times referred to the behavioral traits of the migratory locusts reported in the previous studies. We added more details in the Materials and methods section to include the rationale for the different testing times.

We added “The behavioral change from solitarious to gregarious phase is much slower than the change from gregarious to solitarious phase²³. Dopamine fluctuation and receptor expression significantly changed in the brains of solitarious locusts after crowding for 4 h, and changed in the brains of the gregarious locusts after 1 h of isolation¹⁹. In the present study, the duration of pharmacology intervention is determined as 1 h in the gregarious locusts and 4 h in the solitarious locusts according to the previous study¹⁹.” in Line 481-486.

In Fig 4E (Fig 3 in Ms) the effect of Phf2011 knockdown on behavior was tested in

S animals instead of *G* animals. If the rationale is that miR-9a targets *Phf2011*, then the behavior assay should be conducted in *G* animals to test loss of preference on target knockdown. I assume this was done because relative mRNA levels of *Phf2011* were higher in *S* vs *G* brains. This overlooks the possibility of transient regulation of *Phf2011* by miR-9a independent of other mechanisms which might serve to regulate *Phf2011* in brains of long term *S* or *G* animals.

The suggestion is very reasonable. To clarify this, we detected the behavior of gregarious locust after RNAi knockdown of *phf2011*. However, *phf2011* knockdown did not change the olfactory preference of the gregarious locusts (Figure S7). These results suggest that *phf2011* is not involved in locust olfactory behavior, although it is a target of miR-9a. We speculate *phf2011* mediates the function of miR-9a in other behaviors or phenotypes.

We added “In addition, we detected the olfactory responses of gregarious locusts after *phf2011* knockdown ($t = 3.818$, $P = 0.002$) (Figure S7A), and we found that *phf2011* knockdown did not change the olfactory preference of gregarious locusts ($G_2 = 0.23$, $P = 0.892$) (Figure S7B). These results indicate that *ac2*, but not *phf2011*, is involved in locust olfactory preference.” in Line 227-231, and Figure S7 in the supplemental materials.

Figure S7

Figure S7. Knockdown of *phf2011* did not change the olfactory preference of the gregarious locusts. (A) The mRNA level of *phf2011* after injection of dsPhf2011 in the brain of the gregarious locusts. The mRNA level of *phf2011* are presented as the mean \pm SEM ($n = 8$). (B) The olfactory responses of the gregarious locusts after *phf2011* RNAi knockdown ($n = 45$ and 41). The asterisks outside the strip indicate the significant difference between controls and the treatments through Student's *t*-test and *G*-test for independence. n.s., not significant; **, $P < 0.01$.

Minor details

Figure panels 3 and 4 have been interchanged.

We corrected Figure 3 and 4 in the revised version.

In Fig 8A decrease in La band intensity on SFK treatment is difficult to visualize.

We enlarged the part around 55kD of the pictures to make the La band easier to be visualized. The details are presented in Figure 8.

Primer sequence for Pri-miR-9a qPCR assays is not listed.

We added the primer sequence of Pri-miR-9a for qPCR assays in Table S6 of supplemental materials.

References to establish specificity of agonist/antagonists used are missing.

Thanks. We added the references for Dop1 antagonist SCH23390 in Line 138, Dop1 agonist SKF38393 in Line 140, AC2 antagonist SKF83566 in Line 236.

Line 167 typing error - dsGFP should be dsDop1.

We correct dsGFP to dsDop1 in Line 168.

Since, in mammals, bound La serves to protect pre-miRs from nucleases, it is interesting that there is no change in pre-miR-9a levels on La knockdown and/or Dop1 activation by SKF. This suggests that there must be an alternative mechanism by which La-binding increases mature microRNA levels.

The speculation is very reasonable, and we agree with that. Liang et al. (2013) found the cooperative relationships between La/SSB and Dicer in human cancers and they suggested that the comparable binding affinity allowed pre-miRNAs to be easily transferred from La/SSB to the Dicer complex for processing into mature miRNAs. Thus, we speculate that La protein not only protects pre-miRNAs from nucleases, but also facilitates the biogenesis of mature miRNAs with other regulators such as Dicer. The mechanism by which La-binding increases mature miRNAs levels need to be further investigated in future work.

We discussed this point in the Discussion section. We added “La protein has been proved to promote miRNA biogenesis by stabilizing pre-miRNAs from nuclease-mediated decay in HeLa cells⁴⁴. However, in the migratory locusts, the expression level of pre-miR-9a did not change after La RNAi knockdown or Dop1 activation. These results suggested that La protein may not maintain the level of pre-miR-9a through preventing nuclease-mediated decay. The results in human breast cancer cells suggested that cooperative relationships between La/SSB and Dicer allowed pre-miRNAs to be easily transferred from La/SSB to the Dicer complex for maturation processing⁴⁴. Thus, La protein possibly facilitates the biogenesis of miR-9a from pre-miR-9a with other cooperative regulators, such as Dicer. The molecular mechanism by which La protein facilitates miR-9a maturation need to be further investigated.” in Line 415-425.

Reviewer #3 (Remarks to the Author):

The paper addresses interesting issues and reports interesting results. Unfortunately, the work is not well presented and there are several important aspects that need to be addressed before the work can be published.

- 1) The main behavioral assay, upon which the entire paper depends, is a Y-maze to test preference between no odor and volatiles from gregarious locusts. Unfortunately, no information is given on whether the observed differences really are due to differences in odor preference rather than differences in arousal/inquisitiveness.***

Thanks for the comments. The olfactory behavior in Y-maze has been well established in the locust studies (Guo et al., 2011; Ma et al., 2015; Wang et al., 2015). The diagram of the Y-maze is placed in a room specific for olfactory behavioral assay. The Y-tube is placed in an enclosed and sound insulated cage with a transparent window, to avoid the disturbance from the observer or other ones during behavioral assay. In the previous and present studies, we found that most locusts did not show exploratory behavior in the Y-tube like mice, and did not changed their choice in a period of time after they chose one arm of Y-tube. In addition, the gregarious and solitary locusts did not showed significant preference in Y-tube when used pure air as a stimulus in both arms.

Figure S1. The diagram of the Y-maze as used for testing olfactory preferences of the locusts in the present study.

The olfactory responses of gregarious and solitary locusts in Y-maze when used air as a stimulus.

We added more information about the diagram of Y-maze in Materials and methods section to avoid the doubt that the observed differences are due to differences in arousal/inquisitiveness.

We added the above figure to the supplemental materials (Figure S1) and added “The olfactory behavior was examined in a Y-maze. The diagram of the Y-maze (Figure S1) is placed in a room specific for olfactory behavioral assay. The Y-tube is placed in an enclosed and sound insulated cage with a transparent window, to avoid the disturbance from the observer or other ones during behavioral assay. Each locust was observed for 4 min and examined only once. Whenever the individual locust moved more than 4 cm into the volatile or air arm within 4 min, this individual locust was recorded and considered as the first choice for the respective arm.” in Line 445-451.

2) *More information should be given on the source of volatiles and their concentrations*

Thanks for the suggestions. The odorants used were the volatiles emitted from the body and feces of 40 fourth-stadium gregarious locusts. The components and corresponding concentrations are analyzed in the previous study from our group (Wei et al., 2016). To clarify this, we cited this reference in the Materials and methods section.

We added “The odorants used were the volatiles emitted from the body and feces of 40 fourth-stadium gregarious locusts. The components and corresponding concentrations are analyzed and listed in the previous study²⁴. The volatiles were delivered to either arm to eliminate possible spatial bias. The air flow was set at 300 mL/min.” in Line 452-455.

3) *Is the protocerebrum not more associated more with vision than olfaction?*

The protocerebrum includes two main parts, mushroom body (Kenyon cells,

calyces, pedunculus, lobes) and central complex. Recent studies in locusts revealed the association of mushroom body with olfaction (Cassenaer and Laurent, 2007; Laurent and Davidowitz, 1994; Leitch and Laurent, 1996; Orive et al., 2002; Shen et al., 2013), and the association of central complex with vision (Heinze and Homberg, 2009; Homberg et al., 2003; Trager et al., 2008). In the present study, we found that Dop1, miR-9a, and AC2 mainly located in the Kenyon Cells, which is intrinsic cells of mushroom body. Thus, we suggested that their location is associated with olfactory behavior.

To avoid the confusion, we revised the statement in the main text and legends and described the results more accurately as “Moreover, double-labeling immunohistochemistry in the brain showed that AC2 and Dop1 co-localized in the Kenyon cells of the locust mushroom body, suggesting the functional association of AC2 and Dop1 with olfactory processing (Figures 6E).” in Line 272-275, “The squares specifically indicate the areas where miR-9a and *ac2* were localized in the Kenyon cells of mushroom body.” in Line 1061-1063, “The squares specifically indicate the areas where Dop1 and AC2 were localized in the Kenyon cells of the locust mushroom body.” in 1105-1106.

There should be a detailed section on bioinformatic methods with information on stringency parameters, etc. (e.g. mirDeep, miRanda).

Thanks for the comments. We added more details in in the Materials and methods section. We added “We employed the algorithms miRanda and RNAhybrid to predicate the potential target genes of miR-9a in the gene database from locust transcriptome. The alignment score of miRanda was restricted to 140, and other parameters are default. The parameters of RNAhybrid is set as 1 hits per target, free energy threshold is -17 kcal/mol, helix constraint is 2 to 6, max bulge loop length is 2, max internal loop length is 6, no G:U in seed is included. The predicted results are listed in Table S1, which include the accession number of genes in the transcriptome, gene lengths, position of alignment.” in Line 601-607.

5) What is the reason for the differences in timing (48hr, 72 hr, 1hr, 4hr...). No explanation is provided.

We chose different timing for different treatments dependent on their significant effects in gene expressions and behavioral changes, according to the previous studies (Ma et al., 2011; Guo et al., 2011; Guo et al., 2015; Ma et al., 2015; Yang et al., 2014). We added more details to explain the reason for the different timing in different treatments in the Materials and methods section.

For RNAi experiments, we found that injection of dsRNA for 72 h can significantly downregulated gene expressions and induced behavioral changes (Ma et al., 2011; Guo et al., 2011; Guo et al., 2015; Ma et al., 2015). Thus, we chose 72 h for RNAi experiments in the present studies. We added “The duration for RNAi is determined according to the previous studies^{19, 20, 23}.” in Line 517-518.

For agomir and antagomir injection, we referred to the previous study (Yang et

al., 2014), and chose 48 h for these experiments. We added “The duration of agomir and antagomir treatment is determined according to the previous study²¹.” in Line 641-642.

For pharmacology experiments, we referred to the previous study (Guo et al., 2015), and chose 1 h and 4 h for solitarious and gregarious locusts. We added “The behavioral change from solitarious to gregarious phase is much slower than the change from gregarious to solitarious phase²³. Dopamine fluctuation and receptor expression significantly changed in the brains of solitarious locusts after crowding for 4 h, and changed in the brains of the gregarious locusts after 1 h of isolation¹⁹. In the present study, the duration of pharmacology intervention is determined as 1 h in the gregarious locusts and 4 h in the solitarious locusts according to the previous study¹⁹.” in Line 481-486.

6) *Would be good to also show data points rather than bars +/- sem on the graphs*

The suggestion is very helpful. We changed the graphs of Dop1, AC2, and miR-9a quantification by qPCR and western blot. The details are presented in Figures 1, 3, 5, and 6.

7) *The behavioral experiments should be more clearly describe in the text. For example, it should be made clear that experiments are done with a Y-maze.*

The comments are very helpful. We clarified that the behavioral experiments are all done with a Y-maze in the text. We added “We assessed olfactory preferences by giving each locust a single choice between gregarious volatiles and air in a Y-maze (Figure S1) throughout this study.” in Line 130-131.

8) *Samples sizes are not clear. It is mentioned that "Every experiment encompassed at least six replicates, and every replicate contained at least eight insects." But then for their statistics it is mentioned that $n > 40$. This raises the question of what are the “replicates” and whether there are issues of pseudoreplication. Also why not using fisher’s exact instead of the G-test?*

We tested the choice of locusts in the Y-tube individually, and each locust is a replicate. For statistical analysis, we pooled all insects tested and calculated the significance using *G*-test for independence. The data of behavioral assay is presented as $p \pm SE$. The standard error (SE) of locusts’ volatile preference was calculated as $\sqrt{p(1-p)/n}$, where p is the proportion of locusts that were attracted by the volatiles and n is the number of locusts that were tested. We added the exact number of locusts for each treatment in the figure legends.

G-test is similar to Chi square test, and these two tests are widely used in Y-maze behavioral assay. We used *G*-test for the behavioral assay analysis referred to the previous studies on locust olfactory behaviors (Simoes et al., 2013; Ma et al., 2015). We added more information in the section of Statistical analysis. We added “The

statistical analysis of olfactory preference was referred to the work on locust aversive learning⁶⁰ and olfactory attraction/repulsion²². The olfactory responses between the control and treatment were analyzed using *G*-test for independence, and quantities of all locusts tested were included for this analysis. The data of behavioral assay is presented as $p \pm SE$. The standard error (SE) of locusts' volatile preference was calculated as $\sqrt{p(1-p)/n}$, where p is the proportion of locusts that were attracted by the volatiles and n is the number of locusts that were tested. Differences were considered significant at $P < 0.05$." in Line 777-784.

We also analyzed our data again using Fisher's exact test, and we found that the statistical results analyzed by Fisher's exact test are similar to our results in the manuscript. We listed the statistical results from *G*-test, Chi-Square test and Fisher exact test in the Table below.

Experiments	G -test	Chi-Square test	Fisher exact test
S vs. G	$G_2=14.64, P < 0.001$	$\chi^2=16.470 P < 0.001$	$P < 0.001$
G + dsDop1	$G_2= 27.74, P < 0.001$	$\chi^2= 21.595 P < 0.001$	$P < 0.001$
G + SCH23390	$G_2= 66.63, P < 0.001$	$\chi^2= 9.463 P = 0.002$	$P = 0.003$
S + SKF38393	$G_2= 18.41, P < 0.001$	$\chi^2= 17.486 P < 0.001$	$P < 0.001$
G + agomir-9a	$G_2= 12.714, P = 0.002$	$\chi^2= 6.954 P = 0.008$	$P = 0.011$
S + antagomir-9a	$G_2= 7.03, P = 0.029$	$\chi^2= 5.166 P = 0.023$	$P = 0.025$
miR-9a rescue	$G_2= 7.87, P = 0.019$	$\chi^2= 4.270 P = 0.039$	$P = 0.040$
G + dsAc2	$G_2= 9.16, P = 0.010$	$\chi^2= 7.635 P = 0.006$	$P = 0.007$
S + Forskolin	$G_2= 21.84, P < 0.001$	$\chi^2=10.434 P = 0.001$	$P = 0.002$
G + SKF83566	$G_2= 11.08, P = 0.004$	$\chi^2= 6.349 P = 0.012$	$P = 0.014$
AC2 rescue	$G_2= 8.37, P = 0.015$	$\chi^2= 5.717 P = 0.017$	$P = 0.030$

The "replicates" here means "groups" during injection operation. We found that injection operation took about 30-60 seconds for each individual. For the short timing, such as 1 h or 4 h, the accurate action time for each locust will be different if we injected all the locusts (more than 40 for each treatment) once. So we separated the

treatment and control locusts to several groups. To avoid the confusion, we revised this part as “We assessed olfactory preferences by giving each locust a single choice between gregarious volatiles and air in a Y-maze, and each locust is a replicate. For statistical analysis, we pooled all insects tested and calculated the significance using G-test for independence. To quantify the choice behavior, we directly calculated the percentage of locusts that selected the volatile arm to all tested locusts.” in Line 456-460.

9) *How many animals brains are used for analyses in Fig 1? Are these pooled samples?*

Yes, we pooled 8-10 brains for one sample, and we used 8 and 6 samples for qPCR and western bolt, respectively. We added these details in Materials and methods section.

We added “Six samples of locust brains (8-10 individuals/sample) were collected and homogenized in Trizol reagent (Life Technology) and protein for Western blot analysis was extracted following manufacturers’ instructions.” in Line 462-464, and “Eight samples of locust brains (8-10 individuals/sample) were collected and homogenized in Trizol reagent (Life Technology) and the total RNA was extracted following manufacturers’ instructions.” in Line 520-522.

10) *How are clusters determined? Never mentioned.*

Thanks for the comments. We added more details on the bioinformatics analysis in Materials and methods section. We added “Unsupervised hierarchical clustering was performed using Clustal 3.0⁵⁶, which employs uncentered Pearson correlation and average linkage; the results are presented by Java Treeview software⁵⁷. We determined the clusters a, b, c, and d according to the expression patterns of the miRNAs in SKF and dsDop1 groups. Cluster a includes the miRNAs, which are down-regulated in SKF group and up-regulated in dsDop1 group. Cluster b includes the miRNAs down-regulated in both SKF and dsDop1 groups. Cluster c includes the miRNAs up-regulated in both SKF and dsDop1 groups. Cluster d includes the miRNAs up-regulated in SKF group and down-regulated in dsDop1 group. The thermal graph includes the miRNAs showed significant change in either group or both.” in Line 566-575.

11) *The agomir-NC is not explained in the legend or text.*

We explained the agomir-NC in the Materials and methods section, “The sequence of a *Caenorhabditis elegans* miRNA, cel-miR-67-3p (5'-UCA CAA CCU CCU AGA AAG AGU AGA-3'), was used as a negative control.” in Line 631-633.

We added “...group injected with agomir-NC (cel-miR-67-3p used as the negative control)” in Line 178 of text, and “NC, negative control” in the legends.

12) *The mutant mir9a is not explained in the legend (what is mutated? we don't*

know.) or methods.

The mutant sequence (MT) of the target genes is explained in Figure S5. Some nucleotides in the region of target genes complementary to miR-9a, were mutated to other nucleotides, such as A to U, and C to G. The mutation abolished the interaction of miR-9a and the potential target genes. We described this “The mutations in the binding site of target genes (Figure S5) abolished the suppression effect of miR-9a on the reporters with target sites from *phf201l* CDS, *ac2* CDS, and 3'UTR ($t = 5.050$, $P < 0.001$ for *phf201l* CDS; $t = 4.906$, $P < 0.001$ for *ac2* CDS; $t = 3.611$, $P = 0.004$ for *ac2* 3'UTR) (Figure 4D).” in Line 216-219.

To describe these mutations more clearly, we added “Mutagenesis PCR was performed at the miR-9a target sites by using the Fast Mutagenesis System (TransGen). The point mutations in the putative target sites (WT, orange) were engineered in the region complementary to the miR-9a seed sequence (red). Those point mutations of MT sequences are indicated in blue.” in Line 589-592.

Figure S5. The mutation sites in the putative target genes. (A) The mutation sites in 3'UTR region of *amd1* gene. (B) The mutation sites in CDS region of *phf2011* gene. (C) The mutation sites in CDS region of *ac2* gene. (D) The mutation sites in 3'-UTR region of *ac2* gene. The point mutations in the putative target sites (WT, orange) were engineered in the region complementary to the miR-9a seed sequence (red). Those point mutations of MT sequences are indicated in blue.

13) The entire bit on cytoplasm vs. nucleus doesn't seem useful. I don't see the value to the story. Also they never describe the methods for the fractionation.

The biogenesis of miRNAs is a complex process from nucleus to cytoplasm as

presented in the figure.

Editorial Note: This image has been redacted due to copyright claims.

(Gregory et al., 2005)

To determine which step in miR-9a biogenesis is influenced by Dop1, we tested the expression of pri-miR-9a and pre-miR-9a in total cells of the brains first and found that Dop1 did not affect their expressions. However, we can't directly conclude that Dop1 only influence the maturation process from pre-miR-9a to miR-9a if the specific quantification of pri-miR-9a in nucleus and the specific quantification of pre-miR-9a in nucleus and cytoplasm are not conducted. Thus, to ensure that Dop1 only influence the maturation of miR-9a, we added the specific quantification of pri-miR-9a and pre-miR-9a in different part of the cells.

To describe the methods for the fractionation, we added “The separation of nuclear and cytoplasmic fractions are referred to the previous study³² with slight modifications. The brains were homogenized in cold PBS containing 0.2% Nonidet P-40. The lysate was centrifuged at $30 \times g$ for 2 min at 4 °C to remove the insoluble fragment of tissue. Then, the supernatant was centrifuged at $1500 \times g$ for 15 min at 4 °C. The nuclei were in the pellet whereas the cytoplasm remained in the supernatant. The nuclei were resuspended two times in PBS to remove the residual cytoplasm. The cytoplasmic fraction was centrifuged at $2,000 \times g$ for 10 min at 4 °C to remove the residual nuclei. The nuclei and cytoplasm were added 1 ml Trizol and the total RNA in nuclear and cytoplasmic fractions are extracted as described above. U6 was used as a nuclear marker and 18S rRNA as a cytoplasmic marker to verify the fraction quality.” in Line 701-711.

14) *The authors never describe the mass spec methods or explain their “identify score”.*

Thanks for the comments. We added the details about mass spec methods and explain the identify score in the Materials and methods section. We added the explanation for identify score in Table S4.

We added “The bands in SDS-PAGE gels are accurately cut and collected in an

Eppendorf tube for mass spectrometry analysis in BPI. The proteins were digested to peptides and centrifuged at 20000g for 15min. After that, 10 μ l supernatant was loaded on a ultimate3000 nanoLC (Dionex) by the autosampler onto a 2cm C18 trap column. Then, the peptides were eluted onto a 10cm analytical C18 column (inner diameter 75 μ m) packed in-house. The separated peptides were subjected to nanoelectrospray ionization followed by tandem mass spectrometry (MS/MS) in a Q Exactive (Thermo scientific) coupled online to the HPLC. Peptides were selected for MS/MS using (HCD) operating mode with a normalized collision energy setting of 28%. Intact peptides were detected in the Orbitrap at a resolution of 70000 (first order) and 17500 (second order). The score for each peptide indicates ions score that is $-10\log(P)$, where P is the probability that the observed match is a random event. In the present study, individual ions score > 36 indicates identity or extensive homology ($p < 0.05$), and the peptides with score > 36 are potential candidates to be further tested using Western blot.” in Line 749-762.

We added “* Score indicates ions score that is $-10\log(P)$, where P is the probability that the observed match is a random event. Individual ions score > 36 indicate identity or extensive homology ($P < 0.05$).” in Table S4 in the supplemental materials.

15) The ac2 western blot are a bit messy, which gives doubt to the quality of the densitometry quantification. All the more reason to see the points rather than a bar plot.

Thanks for the comments. We enlarged the pictures to make it clearer to be visualized, and changed the graphs from bars to points. The details are presented in Figures 5 and 6.

16) Seems that figures 3 and 4 are swapped.

We corrected Figure 3 and Figure 4.

Reviewers' comments:

Reviewer #2 (Remarks to the Author):

The authors addressed all the reviewers' concerns in a fairly transparent and detailed manner.

There were a few things in rebuttal to my comments that are still unclear. These are extremely minor though.

- Details of the locust transcriptome used for target prediction (species, version number if applicable) were not listed.
- Table S1 would be better with gene names and associated GO categories from the thermal graph in Figure 4A. Target prediction score (if applicable) would be a nice addition, although I can see why that's not very useful information.
- Typing error in line 601, predicate should be predict.

Overall , I thought they did a good job of responding to every point and all the other reviewers' comments were fair.

Reviewer #3 (Remarks to the Author):

The authors did a good job in addressing the reviewers' comments (especially bioinformatics, and generally improved clarity). There is however one point that needs to be clarified. The authors claim that "Dop1 enhances conspecific olfactory attraction..." . To make this claim would require to show that attraction is *specific* to conspecifics and not generalised attraction to an olfactory stimulus. In that respect it is important to note that dopamine has been associated with general arousal in Drosophila (Ueno 2012 <http://www.nature.com/neuro/journal/v15/n11/full/nn.3238.html>, Kume 2005...). So the question is what happens when locusts are presented with other odors (not air, not volatiles of other locusts)? Do they show the same responses as they do to the volatiles of conspecifics? For example, do the locusts respond similarly to the volatiles of another insect species ? This is an important piece of information that is required to evaluate the claims made in this paper.

Adria Leboeuf and Laurent Keller

Reviewer #1 (Remarks to the Author):

The revisions that authors made in response to my previous comments No.1, 4 and 5 clarified the issues, while there are still some problematic points, which affect the main conclusion of this work:

1) Although specific quantification of mature miR-9a is key to the main conclusion of this paper, it is still confusing how the authors distinguished mature miR-9a without detecting pre-miR-9a. The primers designed for detecting miR-9a should anneal pre-miR-9a as well (Figure S12). If so, the authors need to moderate their claim. Clarify.

In order to answer this question, herein we make more detailed explanations. To make sure the specific quantification of mature miR-9a, we sequenced the amplicons of qPCR samples. We found that the sequencing results were unique in the different samples used in qPCR. The kit for reverse transcribing miRNAs adds Adaptor -Poly(A) to the 3'-terminal of miRNA. Thus, the sequencing results are combined sequences of mature miR-9a-Ploy (A)-Adaptor. The sequence of miR-9a is located in 5'-terminal of pre-miR-9a. If pre-miR-9a is nonspecifically amplified, the sequencing results should include sequences of mature miR-9a-antisense sequence -Ploy (A)-Adaptor. Thus, the calculation of Ct value in qPCR experiments is dependent on the amplification of mature miR-9a, but not that of pre-miR-9a. The original sequencing result (Life Technology) is listed as the following:

To clarify this point, we revised Figure S12 (S13 in the revised version), and added more details. We labelled the 5' and 3'-terminal of pri-miR-9a, pre-miR-9a, and miR-9a in Figure S13 in the revised version. We also labelled the 5' and 3'-terminal of the predicted stem-loop structure of pre-miR-9a and the position of miR-9a in pre-miR-9a in Figure 7A. We added more details in Materials and methods section as “The PCR amplifications of miR-9a are sequenced to verify pre-miR-9a are rarely amplified in these samples.” in Line 575-576.

2) The authors explained the discrepancy of the results in Figure2B and

Figure 2C&D with the methodological difference of RNA-seq and qPCR. However, these methods gave the qualitatively opposite results for genes like miR210-5p, miR263a-5p, and miR190. The authors should mention this discrepancy. Also, they should clarify that the conclusion on the changed expression of miR-9a in response to Dop1/SKF is mainly based on qPCR results.

Thanks for these comments. We revised this part of results and added more details. We added “We applied quantitative polymerase chain reaction (qPCR) to validate the expression patterns of the miRNAs showing the opposite changes (clusters a and d) after the activation and inhibition of Dop1. The expression of miR-9a and miR-2765 significantly decreased ($t = 2.503$, $P = 0.026$ for miR-9a; $t = 2.498$, $P = 0.028$ for miR-2765), whereas bantam-3p and miR-210-5P significantly increased ($t = 2.602$, $P = 0.023$ for bantam-3p; $t = 2.624$, $P = 0.020$ for miR-210-5P) in the solitarious brains after SKF injection (Figure 2C). In the gregarious brains injected with dsDop1, the expressions of miR-133 and miR-278-3p decreased significantly ($t = 3.217$, $P = 0.010$ for miR-133; $t = 2.308$, $P = 0.037$ for miR-278-3p), whereas miR-9a, miR-193, and miR-315-5P increased significantly ($t = 4.083$, $P = 0.002$ for miR-9a; $t = 2.501$, $P = 0.029$ for miR-193; $t = 2.650$, $P = 0.023$ for miR-315-5P) (Figure 2D). The expression levels of miR-210-5P, miR-133, and miR-278-3p detected by RNA-seq and qPCR showed discrepancy. Based on the qPCR results, miR-9a showed significant and opposite changes in SKF and dsDop1 injection groups. Thus, miR-9a significantly responds to Dop1 administration in the brain of the migratory locust.” in Line 174-189.

Reviewer #2 (Remarks to the Author):

The authors addressed all the reviewers' concerns in a fairly transparent and detailed manner. There were a few things in rebuttal to my comments that are still unclear. These are extremely minor though.

Details of the locust transcriptome used for target prediction (species, version number if applicable) were not listed.

Thanks. The sequences used for target prediction is from the brain transcriptome of the migratory locust, *Locusta migratoria*. The transcriptome data has been uploaded to NCBI, and the accession number is SRP116073. The data will be released several months later. We added these details in the text as “We employed the algorithms miRanda³⁸ and RNAhybrid³⁹ to predict the potential target genes of miR-9a in the gene database from locust brain transcriptome (*Locusta migratoria*) (Table S1).” in Line 213-215. The accession number is also listed in the Materials and methods Section in Line 651.

Table S1 would be better with gene names and associated GO categories from the thermal graph in Figure 4A. Target prediction score (if applicable) would be a nice addition, although I can see why that's not very useful information.

We added the gene names and associated GO categories into the Table S1, and added the prediction score in Table S1 and S2. The details are presented in Table S1 and S2. We added more details as “The predicted results are listed in Table S1, which include the accession number of genes in the transcriptome, gene lengths, position of alignment, predict score, gene names, and GO categories.” in Line 634-636.

Typing error in line 601, predicate should be predict.

Thanks. We corrected the typing error form “predicate” to “predict” in Line 629.

Overall, I thought they did a good job of responding to every point and all the other reviewers' comments were fair.

Reviewer #3 (Remarks to the Author):

*The authors did a good job in addressing the reviewers' comments (especially bioinformatics, and generally improved clarity). There is however one point that needs to be clarified. The authors claim that "Dop1 enhances conspecific olfactory attraction...". To make this claim would require to show that attraction is *specific* to conspecifics and not generalised attraction to an olfactory stimulus. In that respect it is important to note that dopamine has been associated with general arousal in *Drosophila* (Ueno 2012 <http://www.nature.com/neuro/journal/v15/n11/full/nn.3238.html>, Kume 2005...). So the question is what happens when locusts are presented with other odors (not air, not volatiles of other locusts)? Do they show the same responses as they do to the volatiles of conspecifics? For example, do the locusts respond similarly to the volatiles of another insect species? This is an important piece of information that is required to evaluate the claims made in this paper.*

This is an interesting question. We tested the responses of the gregarious locusts to the volatiles of several insect species, including *Solenopsis invicta*, *Acheta domesticus*, *Blattella germanica*, *Apis mellifera*, and *Helicoverpa armigera*, respectively. When the migratory locusts are presented with the volatiles and air as control, the locusts showed repulsion to the volatile of red fire ant (*Solenopsis invicta*), but no preference to the volatiles from other insect species. On the other hand, the migratory locusts preferred the volatiles of gregarious locusts to the volatiles of other insects when the volatiles of gregarious locusts are presented in the other arm of Y-maze. These results indicate the gregarious locusts specially prefer the volatiles released by conspecific gregarious locusts. The results are listed below.

Figure S2

Figure S2. The gregarious locusts are specifically attracted to the conspecific gregarious volatiles. (A) The olfactory responses of gregarious locusts to air and the volatiles of *Solenopsis invicta* (n = 41), *Acheta domesticus* (n = 49), *Blattella germanica* (n = 41), *Apis mellifera* (n = 44), *Helicoverpa armigera* (n = 50). (B) The olfactory responses of gregarious locusts to gregarious volatiles and the volatiles of *Solenopsis invicta* (n = 41), *Acheta domesticus* (n = 47), *Blattella germanica* (n = 41), *Apis mellifera* (n = 42), *Helicoverpa armigera* (n = 42). The asterisks (* and **) inside the strip indicate the significant differences of comparing individual numbers in each arm after G-test for goodness-of-fit (*, $P < 0.05$; **, $P < 0.01$; n.s., not significant).

We added these results in the text and Figure S2 in the supplemental materials. We added “To determine whether the gregarious volatiles are the specific olfactory stimuli in the locust attraction, we tested the responses of gregarious locusts to the volatiles from other insect species. The gregarious locusts showed repulsion to the volatiles of *Solenopsis invicta* ($G_1 = 4.19$, $P = 0.020$), and showed no preference for the volatiles of *Acheta domesticus* ($G_1 = 0.24$, $P = 0.313$), *Blattella germanica* ($G_1 = 0.01$, $P = 0.5$), *Apis mellifera* ($G_1 = 0.01$, $P = 0.5$), and *Helicoverpa armigera* ($G_1 = 1.23$, $P = 0.133$), when the locusts are presented with volatiles and air (Figure S2A). After giving the volatiles of gregarious locusts and other insect species, the gregarious locusts preferred conspecific gregarious volatiles to the volatiles of *Solenopsis invicta* ($G_1 = 13.95$, $P < 0.001$), *Acheta domesticus* ($G_1 = 5.82$, $P = 0.007$), *Blattella germanica* ($G_1 = 3.96$, $P = 0.023$), *Apis mellifera* ($G_1 = 3.96$, $P = 0.023$), and

Helicoverpa armigera ($G_1 = 7.46$, $P = 0.003$) (Figure S2B). Thus, the migratory locust shows specific olfactory attraction to volatiles of conspecific gregarious locusts, but not general to an olfactory stimulus.” in Line 134-147.

We added “To assess the specificity of attraction in migratory locusts to the gregarious volatiles, the volatiles from other species (*Solenopsis invicta*, *Acheta domesticus*, *Blattella germanica*, *Apis mellifera*, and *Helicoverpa armigera*) were presented in one arm of a Y-maze. The total body weight of other species was the same as that of 40 fourth-stadium gregarious locusts to eliminate the individual differences among species. The behavioral assay was performed as described above.” in Line 482-487.

We added “In the statistical analysis of olfactory preference, G -test for goodness-of-fit was used to determine the significance of the divergence from an expected 50% decision for the volatile arm or air control arm, and the counts of migratory locusts preferring the volatile or air control were used for this analysis.” in Line 808-811.